

# Retrieval and parametrisation of sea-ice bulk density from airborne multi-sensor measurements

Arttu Jutila[1], Stefan Hendricks[1], Robert Ricker[1,a], Luisa von Albedyll[1], Thomas Krumpen[1], and Christian Haas[1,2]

[1]Alfred Wegener Institute, Helmholtz Centre for Polar and Marine Research, Bremerhaven, Germany
[2]Institute of Environmental Physics, University of Bremen, Bremen, Germany
[a]Present address: Technology Department, Norwegian Research Centre, Tromsø, Norway

**Correspondence:** Arttu Jutila (arttu.jutila@awi.de)

**Abstract.** Knowledge of sea-ice thickness and volume depends on freeboard observations from satellite altimeters and in turn on information of snow mass and sea-ice density required for the freeboard-to-thickness conversion. These parameters, especially sea-ice density, are usually based on climatologies constructed from in situ observations made in the 1980s and before while contemporary and representative measurements are lacking. Our aim with this paper is to derive updated sea-ice

bulk density estimates suitable for the present Arctic sea-ice cover and a range of ice types to reduce uncertainties in sea-ice thickness remote sensing. Our sea-ice density measurements are based on over 3000 km of high-resolution collocated airborne sea-ice and snow thickness and freeboard measurements in 2017 and 2019. Sea-ice bulk density is derived assuming isostatic equilibrium for different ice types. Our results show higher average bulk densities for both first-year ice (FYI) and especially multi-year ice (MYI) compared to previous studies. In addition, we find a small difference between deformed and possibly

unconsolidated FYI and younger MYI. We find a negative-exponential relationship between sea-ice bulk density and sea-ice freeboard and apply this parametrisation to one winter of monthly gridded CryoSat-2 sea-ice freeboard data. We discuss the suitability and the impact of the derived FYI and MYI bulk densities for sea-ice thickness retrievals and the uncertainty related to the indirect method of measuring sea-ice bulk density. The results suggest that retrieval algorithms be adapted to changes in sea-ice density and highlight the need of future studies to evaluate the impact of density parametrisation on the full sea-ice

thickness data record.

## 1 Introduction

Sea ice affects the heat, moisture, and energy exchange between the ocean and the atmosphere, therefore monitoring the state of sea ice is crucial for understanding the current climate, how it may evolve, and what its impact may be (e.g., Stroeve and Notz, 2018). Observing sea-ice thickness and volume over decadal periods relies on freeboard measurements by satellite laser

and radar altimeters. The conversion of freeboard to sea-ice thickness requires information of snow mass as well as the density of the sea-ice layer. Observations of both input parameters are sparse and the unknown spatial and temporal variability and trends of snow mass and sea-ice density directly translate into the uncertainty of the sea-ice thickness data record (Giles et al., 2007; Kwok, 2010; Zygmuntowska et al., 2014). The current uncertainties related to sea-ice thickness and volume retrievals



are sometimes deemed too large for modelling comparisons (SIMIP Community, 2020), and therefore improved accuracy of

sea-ice thickness is highly desired (Duchossois et al., 2018). Coming to the era of satellite altimetry, starting with the European Remote Sensing satellite ERS-1 mission of the European Space Agency (ESA) in 1993, multi-year ice (MYI, ice that has survived at least two melt seasons) covered about 40 % of the late-winter Arctic and already showed signs of reduction in areal coverage. Since then, Arctic sea ice has undergone rapid change due to the warming climate resulting in a thinner and younger ice cover (Maslanik et al., 2011; Comiso, 2012; Meier et al., 2014; Stroeve and Notz, 2018). Toward the end of 2010s, MYI

continued to decline and constituted barely 10 % of the Arctic sea-ice extent while the relative extents of the thinner first-year ice (FYI, ice that has not undergone a melt season) and second-year ice (SYI, ice that has survived one melt season) have increased (Stroeve and Notz, 2018).

Due to the lack of spatially and temporally representative snow observations, sea-ice thickness retrieval methods are based on, e.g., monthly snow climatologies or modelled reconstructions from reanalysis. Currently the most widely used source for

snow mass information is the snow climatology in Warren et al. (1999) (hereafter W99) using data collected during the Soviet North Pole drifting stations in 1954–1991. However, the stations were located exclusively on MYI and thus introduce a bias. In response to the declining MYI, many satellite data products of Arctic sea-ice thickness are derived by using a modified W99, where the snow depth values are halved on ice that corresponds to FYI (Kurtz and Farrell, 2011; Sallila et al., 2019). In attempts to overcome the mismatch between the pre-1990s climatology and the shift toward FYI-dominated, thinner, and younger Arctic

sea ice, a number of new snow depth products have emerged in recent years. W99 was complemented with data from airborne Sever expeditions covering in particular FYI in the shelf seas of the Eurasian Russian Arctic in late-winter (March–May) 1959–1986. Other approaches have utilised atmospheric reanalysis data to model a reconstruction of snow on Arctic sea ice in varying spatial and temporal resolutions. Snow depth has been derived using brightness temperatures from passive microwave satellites as well as combining dual-altimetry freeboard information from Ku band and Ka band or laser satellite altimeters. Descriptions

of the different snow depth products currently available can be found in the inter-comparison study of Zhou et al. (2021). Some constraints of the current products will be reduced by near real-time dual-altimetry acquisitions, such as the resonance of the CryoSat-2 and ICESat-2 satellite orbits (CRYO2ICE) since July 2020, and future single-platform dual-frequency satellite missions like the Copernicus Polar Ice and Snow Topography Altimeter (CRISTAL) mission by ESA (Kern et al., 2020) with a launch planned in 2027. Similar efforts to bring sea-ice density values up-to-date should be taken as well.

Sea ice is a multi-phase substance consisting of solid ice, liquid brine, and gas (air) bubbles with densities and relative amounts depending on temperature (Timco and Frederking, 1996). Calculated from its molecular structure, density of pure ice is 916.8 $\mathrm{kg\,m^{-3}}$ (Pounder, 1965), while liquid, saline water increases and air inclusions decrease the sea-ice bulk density (weight per unit volume including voids and enclosed water). Through processes like brine expulsion, gravity drainage, and meltwater flushing, sea ice is desalinated over time as pore space previously occupied by highly saline brine is replaced by

sea water and air resulting in decreased density of MYI (Petrich and Eicken, 2017). Several studies have demonstrated the rather large range of values for sea-ice density despite little brine drainage and with differences in respect to the waterline and ice type: above the waterline FYI density is 840–910 $\mathrm{kg\,m^{-3}}$ and MYI density is 720–910 $\mathrm{kg\,m^{-3}}$, whereas below the waterline ice is saturated by sea water and has a density of 900–940 $\mathrm{kg\,m^{-3}}$ less dependent on its age (Timco and Frederking,





1996; Timco and Weeks, 2010; Pustogvar and Kulyakhtin, 2016). Especially with satellite altimetry applications in mind,

Alexandrov et al. (2010) derived a density of $916.7 \pm 35.7 \ \mathrm{kg\,m^{-3}}$ for FYI using the drill-hole data set of airborne Sever expeditions concentrated in the shelf seas of the Eurasian Russian Arctic in the 1980s and a density of $882 \pm 23 \ \mathrm{kg\,m^{-3}}$ for MYI as a weighted average of the layers above and below the waterline using values from literature. The values by Alexandrov et al. (2010) (hereafter A10) are the most commonly used in sea-ice thickness retrieval algorithms (Sallila et al., 2019).

Accurate and representative measurements of sea-ice density using traditional techniques are temporally and spatially lim-

ited. Most of them require coring or cutting out pieces of ice, such as the mass/volume, displacement (submersion), or specific gravity techniques, making them susceptible for inaccuracies through brine drainage and imprecise volume of the samples (Timco and Frederking, 1996). This can be avoided by carefully recording sea-ice thickness and freeboard, e.g., with drill-hole measurements, in addition to the snow depth atop and calculating the sea-ice bulk density assuming isostatic equilibrium and densities of snow and sea water. However, significant error may be introduced locally where sea ice is not isostatically compen-

sated due to lateral stresses, e.g., close to pressure ridges (Timco and Frederking, 1996). Previous parametrisations of sea-ice density include the effective freeboard approach (snow depth converted to ice thickness using their density ratio) by Ackley et al. (1976) using drill-hole measurements from $400 \ \mathrm{m}$ of profile lines on MYI in the Beaufort Sea and a parametrisation based on ice floe thickness by Kovacs (1997) utilising 17 FYI and 4 MYI sea-ice cores from the Beaufort Sea. Neither of these parametrisations have been widely used. Moreover, the multi-phase nature of sea ice is an ongoing challenge for modelling

approaches (Hunke et al., 2011). There is a definite need for evaluating sea-ice density because there is no density climatology available representing the current state of sea ice, nor is it possible to observe density by satellites from space.

Simultaneous, collocated, and preferably single-platform measurements of the key parameters of the entire sea-ice–snow layer covering a wide range of ice types and conditions on regional scales are required to decrease the uncertainties related to the conversion of freeboard to sea-ice thickness. Since 2017, a unique sensor configuration on the Alfred Wegener Institute's

(AWI) IceBird winter campaigns combines airborne laser, radar, and electromagnetic induction sounding instruments making it now possible to measure them on a single platform. In this paper, we present high-resolution data of simultaneous airborne sea-ice thickness, freeboard, and snow depth over late-winter Arctic sea ice from the AWI IceBird campaigns in 2017 and 2019. Observing the locations of the air–snow, snow–ice, and ice–water interfaces in the sea-ice system along survey tracks allows us to estimate sea-ice bulk density that also serves as a consistency check between the sea-ice thickness, freeboard, and

snow depth measurements. We also derive an updated parametrisation of sea-ice bulk density suitable for the present Arctic sea-ice cover including the densities of deformed sea ice which, if unconsolidated, can deviate even more strongly from the density of solid ice.

## 2 Data and methods

### 2.1 Aircraft campaigns

The AWI IceBird program (see reference list for a link to webpage) is a series of airborne campaigns carried out using the institute's two Basler BT-67 research aircraft *Polar5* and *6* (Alfred-Wegener-Institut Helmholtz-Zentrum für Polar- und Meeres-




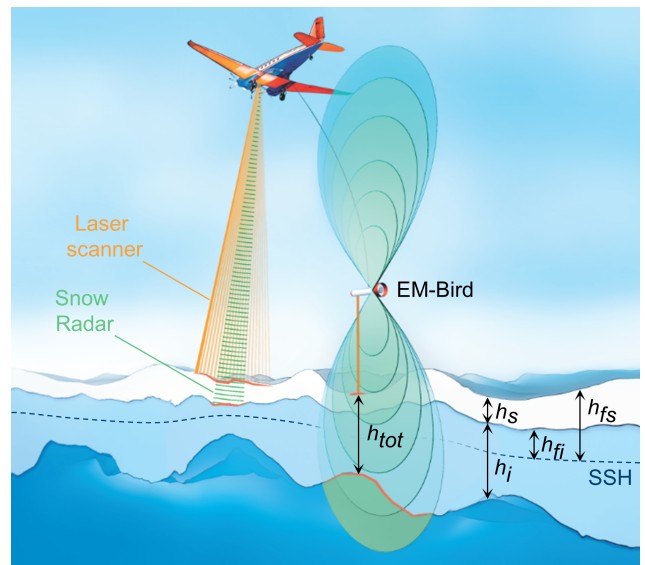

**Figure 1.** Sketch of the IceBird sea-ice campaign setup with the EM-Bird (black text), the laser scanner (orange), and the Snow Radar (green). Different components of sea ice are highlighted in the cross-section: total (ice+snow) thickness ($h_{tot}$), snow depth ($h_s$), sea-ice thickness ($h_i$), sea-ice freeboard ($h_{fi}$), and snow freeboard ($h_{fs}$). SSH stands for local sea-surface height, depicted by the blue dashed line. Adapted with annotations from the graphic by Alfred Wegener Institute / Martin Künsting CC-BY 4.0.

forschung, 2016) to measure Arctic sea ice and its change since 2009. The campaigns operate from airfields extending from Longyearbyen in Svalbard to Utqiagvik (Barrow) in Alaska and coincide closely with the Arctic sea ice summer minimum (August) and winter maximum (April). The primary scientific instrumentation on the aircraft includes an electromagnetic

(EM) induction sounding instrument (EM-Bird) to measure total (i.e., ice+snow) thickness, an airborne laser scanner (ALS) for surface topography and freeboard measurements, a microwave radar to measure snow depth, and an infrared radiation pyrometer to record surface temperature (Fig. 1). We describe each instrument in the following sections. The low altitude of 200 ft ($\approx 60$ m) and slow speed of 110 kn ($\approx 60$ m s$^{-1}$) during the nominal surveys are beneficial for high-resolution data acquisition.

In this study, we used data collected during the IceBird winter campaigns in early April of 2017 and 2019 (Table 1; Fig. 2) utilising the unique data set of simultaneous total thickness, snow freeboard, and snow depth measurements. From 2017, we used measurements from four survey flights that took place over the Beaufort and Chukchi Seas as part of the Polar Airborne Measurements and Arctic Regional Climate Model Simulation Project (PAMARCMiP; Haas et al., 2010; Herber et al., 2012). From 2019, we included five survey flights that covered regions in the Lincoln Sea and the Arctic Ocean in addition to an

overlap with the measurements in the Beaufort Sea in 2017.



**Table 1.** IceBird surveys in 2017 and 2019 in the focus of this study (Fig. 2) together with their total lengths and instrument retrieval rates.

| # | Date | Base | Survey (Region) | Total distance [km] | Retrieval rates [%] | | |
|---|------|------|-----------------|---------------------|--------------------|---|---|
| | | | | | EM-Bird | ALS | Snow Radar |
| 1 | 2 April 2017 | Inuvik, Canada | Beaufort Loop | 416 | 87 | 75 | 81 |
| 2 | 4 April 2017 | Inuvik, Canada | AltiKa track (Beaufort) | 265 | 90 | 99 | 82 |
| 3 | 6 April 2017 | Utqiagvik, USA | Sentinel-3A track (Chukchi) | 463 | 91 | 93 | 79 |
| 4 | 8 April 2017 | Utqiagvik, USA | ULS & UiTSat (Chukchi) | 619 | 81 | 93 | 77 |
| 5 | 2 April 2019 | Eureka, Canada | Nansen Sound, Arctic Ocean | 294 | 90 | 76 | 83 |
| 6 | 5 April 2019 | Eureka, Canada | Lincoln Sea | 189 | 88 | 99 | 78 |
| 7 | 7 April 2019 | Inuvik, Canada | Beaufort Triangle | 470 | 85 | 97 | 54 |
| 8 | 8 April 2019 | Inuvik, Canada | Amundsen Gulf (Beaufort) | 279 | 82 | 97 | 70 |
| 9 | 10 April 2019 | Inuvik, Canada | ICESat-2 track (Beaufort) | 415 | 76 | 98 | 90 |

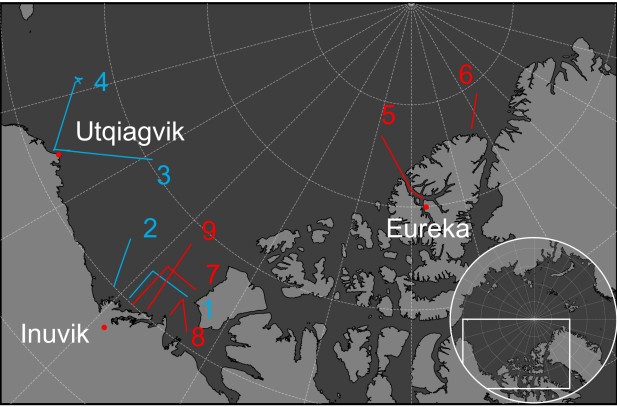

**Figure 2.** IceBird surveys in 2017 (blue) and 2019 (red) in the focus of this study. The numbering corresponds to the individual surveys listed in Table 1.

## 2.2 Sea-ice thickness

We measured total (ice+snow) thickness of sea ice ($h_{tot}$) using the towed EM induction sounding instrument, the EM-Bird, suspended below the aircraft 10–20 m above the sea ice surface (Haas et al., 2009) as illustrated in Fig. 1. The EM-Bird utilises the contrast between resistive snow and ice layers and conductive sea water by transmitting an EM field that induces eddy currents only in the latter (Kovacs and Morey, 1991; Haas et al., 1997). The EM-Bird measures the phase and amplitude of a secondary EM field induced by those eddy currents in relation to the primary field. The phase and amplitude of the secondary field depend on the distance between the instrument and the ice–water interface and decrease negative-exponentially with






increasing distance. Subtracting the instrument height above the surface, measured by an integrated laser altimeter, from the distance to the sea water gives the total thickness (Haas et al., 2009, 2021). Sea-ice thickness ($h_i$) was derived by subtracting

snow depth from total thickness (see Sect. 2.4). The EM-Bird sampling rate was 10 Hz, which translated to 5–6 m point spacing at the nominal survey speed. Approximately every 15–20 minutes brief ascents to more than 100 m were carried out to monitor the sensor drift during post-processing (Haas et al., 2009). Comparison to drill-hole measurements over level ice have indicated an accuracy of 0.1 m, whereas ridge peak thicknesses are generally underestimated by up to 50 % as a result of mass-conserving averaging effects within the approximately 40 m diameter footprint of the instrument (Pfaffling et al., 2007;

Haas et al., 2009). For our analysis, we disregarded measurements of total thickness (i) less than the instrument accuracy of 0.1 m, (ii) where total thickness was less than the mean snow freeboard or snow depth, and (iii) where the surface temperature was above $-5\,°C$ within the footprint of the instrument to avoid open or newly frozen leads with total thickness below the accuracy of the EM-Bird (see Sect. 2.4 and 2.5.1).

## 2.3 Freeboard

The near-infrared (1064 nm), line-scanning Riegl VQ-580 airborne laser scanner (ALS) measured ellipsoidal elevations of ice surfaces with a $60°$ field of view resulting in a swath width approximately equal to the aircraft's altitude above ground (nominally $\approx 60$ m). We obtained freeboard from the ALS data by subtracting the local sea-surface height from the ice surface elevations. The height of the sea surface along the flight track is sporadically observed by the ALS at fractures (leads) of the sea-ice cover and we manually selected the corresponding elevations. We subtracted the mean sea surface (DTU15 MSS; Andersen

et al., 2016) from the surface elevations to remove large scale variations and reduce interpolation errors before interpolating the sea-surface height tie points along the flight track. Subtracting the interpolated sea-surface height from the ice elevations results in snow freeboard ($h_{fs}$), as the elevation measurement of the ALS always includes the snow layer (Fig. 1). Sea-ice freeboard ($h_{fi}$), i.e., the location of the snow–ice interface in relation to the local sea level, was derived by subtracting snow depth from snow freeboard (see Sect. 2.4). We then interpolated the obtained point cloud data of snow freeboard onto a regular

grid with a 0.25 m resolution. Freeboard uncertainties are dominated by the accuracy of the interpolation of the instantaneous sea surface anomaly that depends on the abundance of leads. Therefore, especially regions with packed MYI and low lead density are associated with high uncertainties (Ricker et al., 2016) and we manually masked out such areas. Supervising the along-track interpolation, we estimated an overall uncertainty of 0.1 m. For each total thickness measurement, we calculated the corresponding mean snow freeboard within the EM-Bird footprint (Fig. 3).

## 2.4 Snow depth

We used an ultra-wideband (2–18 GHz), frequency-modulated continuous-wave (FMCW), quad-polarised microwave radar, hereafter Snow Radar, to measure snow depth on sea ice ($h_s$). The radar was developed by the Center of Remote Sensing of Ice Sheets (CReSIS) at the University of Kansas, and similar radars have been operated as part of NASA's Operation IceBridge (OIB) since 2009 (MacGregor et al., 2021). Similar to the ALS, the Snow Radar transmitter and receiver antennae

were mounted under the aft floor of the aircraft looking nadir (Fig. 1). Due to the broad bandwidth of the radar, its range





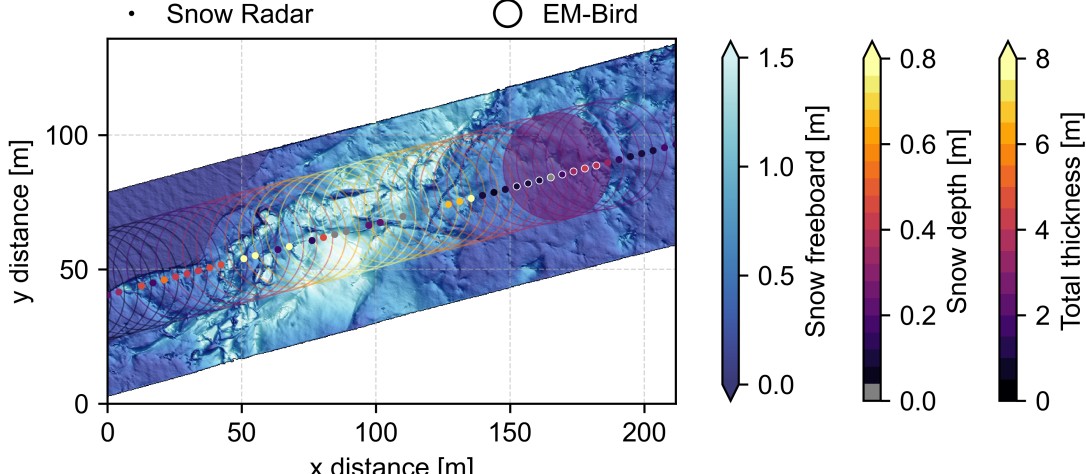

**Figure 3.** Example of the primary data sets: the bluish colours show the gridded 0.25 m resolution swath of snow freeboard ($h_{fs}$) measured by the ALS; the small filled circles are the snow depth estimates ($h_s$) from the Snow Radar, where the diameter of the circle corresponds to the theoretical smooth surface cross-track footprint of the radar; and the large open circles represent the total (ice+snow) thickness measurements ($h_{tot}$) from the EM-Bird in their respective footprint size. One EM-Bird footprint on the right is highlighted (filled transparent colour) to demonstrate the averaging of snow depth estimates (white outlines) and freeboard. A refrozen lead can be seen in the upper left corner of the figure and a pressure ridge in the middle.

resolution was 1.14 cm in snow when assuming a snow density of 300 kg m$^{-3}$. The low altitude and slow speed of the IceBird surveys resulted in an approximately 4–5 m sample spacing and a theoretical smooth surface footprint diameter of only 2.6 m across-track and 1.0 m along-track (Fig. 3). A detailed description of the radar is given in Yan et al. (2017) and Jutila et al. (2021).

To calibrate the raw Snow Radar data, we used a workflow described in Jutila et al. (2021) including coherent noise removal and system impulse deconvolution. Using an open-source python package pySnowRadar (King et al., 2020a), we detected air–snow and snow–ice interfaces with the algorithm by Jutila et al. (2021) that is based on a pulse peakiness approach by Ricker et al. (2014) used in satellite altimetry. Taking into account the decreased wave propagation speed in snow by assuming a snow density, the distance between the identified interfaces determined the snow depth estimate. In post-processing, we filtered

out values which were acquired during the EM-Bird calibration manoeuvres with a simple altitude threshold of 100 m and when the absolute roll or pitch of the aircraft exceeded 5°. Additionally, for each snow depth estimate we calculated a surface topography estimate ($h_{topo}$) as the difference between the 95th and 5th percentile of the ALS surface elevation data within the radar footprint to disregard potentially erroneous snow depth estimates over heavily deformed sea ice using a threshold value of 0.5 m (Jutila et al., 2021).

A validation exercise over level, landfast FYI yielded a mean bias of 0.86 cm and a root-mean-square error (RMSE) of 6.9 cm for the radar-derived snow depth estimates (Jutila et al., 2021). For each total thickness measurement, we calculated the





corresponding mean snow depth requiring at least five valid snow depth estimates within the EM-Bird footprint (corresponds to approximately 50 % of the values; Fig. 3). The averaging reduced the error by a factor of $\sqrt{N}$, where $N$ is the number of averaged estimates. We disregarded averaged snow depth values where the surface temperature within the EM-Bird footprint

was above $-5\,^\circ\mathrm{C}$ and where the mean snow freeboard was less than the mean snow depth (negative sea-ice freeboard) to avoid potentially erroneous snow depth retrievals due to changes in the dielectric properties of snow induced by liquid water (Barber et al., 1995; Kurtz and Farrell, 2011; Kurtz et al., 2013).

## 2.5 Auxiliary data

### 2.5.1 Surface temperature

Surface temperature was acquired using the Heitronics infrared radiation pyrometer KT19.85II that recorded the 9.6–11.5 µm spectral band response of the surface with a sampling rate of 50 Hz resulting in an approximately 1 m sample spacing and 3.1 m diameter footprint at the nominal survey speed and altitude. The manufacturer reported an accuracy of $\pm 0.5\,^\circ\mathrm{C} + 0.7\,\%$ of the temperature difference between the target and the instrument housing. We used the measurements to filter out total thickness and snow depth measurements where the surface temperature was above $-5\,^\circ\mathrm{C}$ (see Sect. 2.2 & 2.4).

### 2.5.2 Sea-ice type

Information of sea-ice type is required for accurate classification the sampled ice. However, no remote sensing product or modelling output is able to match the resolution of the airborne survey data. Therefore, we used a custom sea-ice classification scheme. We started with identifying level and deformed ice following the approach of Rabenstein et al. (2010). Conditions for level ice were met when the along-track total thickness gradient using a three-point Lagrangian interpolator was less than

0.04 and the level ice section extended for at least 100 m. Otherwise, ice was deemed deformed. We then chose the nearest neighbour data point from the coinciding EASE-Grid Sea Ice Age (Version 4) product from the National Snow & Ice Data Center (NSIDC; Tschudi et al., 2019) providing weekly sea-ice age estimates in 12.5 km resolution. Where the NSIDC Sea Ice Age data were not available (landfast ice and close to coasts), we manually assigned the ice type to FYI or MYI (old ice, including SYI) according to the Canadian Ice Service (CIS) regional and weekly ice charts (Canadian Ice Service, 2009).

Finally, we defined the ice type as (i) first-year ice (FYI), if the ice was younger than 1 year according to NSIDC/CIS or the observed ice thickness was below 2 m regardless of its age; (ii) second-year ice (SYI), if the ice had a thickness of 2 m or more and its age was 1–2 years; and (iii) multi-year ice (MYI), if the ice had a thickness of 2 m or more and was at least 2 years old. To account for the spatial and temporal limitations of the NSIDC Sea Ice Age product and the drift of sea ice, we adjusted the ice type classification from FYI to MYI for any ice that indicated an age less than 1 year but was level and thicker than 2 m or,

after along-track averaging over a length scale (see Sect. 2.6), the lower quartile (25th quantile) of the averaged ice thickness values within the length scale was above 2 m.

To support the analysis of the sampled sea ice and to evaluate the indicated sea-ice age in the NSIDC product, we investigated the sea-ice age, pathways, and origin using the Lagrangian drift analysis system ICETrack (Krumpen, 2018; Krumpen et al.,



2020). We split the surveyed sea ice into 25 km along-track segments and tracked them backwards in time in daily increments

utilising a publicly available low-resolution satellite sea ice motion product from the Ocean and Sea Ice Satellite Application

Facility (OSISAF; Lavergne et al., 2010). The tracking was terminated if the trajectory hit the coastline or the edge of landfast

ice. In addition, if the sea-ice concentration, provided by the Center for Satellite Exploitation and Research (CERSAT; Ezraty

et al., 2007), along the backward trajectory dropped below 25 %, we assumed that the ice was formed in that specific location.

To quantify uncertainties of sea-ice trajectories, Krumpen et al. (2019) reconstructed the pathways of 57 drifting buoys. The

authors showed that the deviation between actual and virtual tracks was rather small, $36 \pm 20$ km after 200 days, and considered

to be in an acceptable range.

## 2.6 Sea-ice bulk density

Simultaneous measurements of sea-ice thickness, snow depth, and freeboard enable us to calculate sea-ice bulk density using

the so-called "freeboard and ice thickness technique" (Timco and Frederking, 1996). Archimedes' principle dictates

$$\rho_i h_i + \rho_s h_s = \rho_w \left( h_i - h_{fi} \right) \tag{1}$$

where $\rho$ is density for ice, snow, and sea water denoted with subscripts $i$, $s$, and $w$, respectively. The terms can be rearranged

to solve for $\rho_i$:

$$\rho_i = \rho_w \left( 1 - \frac{h_{fi}}{h_i} \right) - \rho_s \frac{h_s}{h_i}. \tag{2}$$

By substituting measured total thickness and snow depth for sea-ice thickness ($h_i = h_{tot} - h_s$) in addition to snow freeboard

and snow depth for sea-ice freeboard ($h_{fi} = h_{fs} - h_s$), we obtain

$$\rho_i = \rho_w \left( 1 - \frac{h_{fs}}{h_{tot} - h_s} \right) + (\rho_w - \rho_s) \frac{h_s}{h_{tot} - h_s}. \tag{3}$$

To solve Eq. (3), we need to assume values only for the densities of sea water and snow, but their impact on the uncertainty

of sea-ice bulk density is small (see Eq. (4) below). Here we took sea-water density and its uncertainty according to Wadhams

et al. (1992) as $\rho_w = 1024$ kg m$^{-3}$ and $\sigma_{\rho_w} = 0.5$ kg m$^{-3}$, respectively. For snow density in April, when measurements were

carried out, we chose $\rho_s = 300$ kg m$^{-3}$ following Warren et al. (1999) and for the respective uncertainty $\sigma_{\rho_s} = 34$ kg m$^{-3}$

from King et al. (2020b). These values and the uncertainties of the measured variables are summarised in Table 2. Assuming

that the individual uncertainties are uncorrelated, we can derive uncertainty for sea-ice bulk density ($\sigma_{\rho_i}$) using Gaussian error



**Table 2.** Summary of the key variables and their assumed values, uncertainties, and resolutions.

| Variable | Unit | Value | Uncertainty ($\sigma$) | Resolution |
|---|---|---|---|---|
| Total thickness ($h_{tot}$) | [m] | | $0.1^a$ | 5–6 m spacing, 40 m footprint |
| Snow freeboard ($h_{fs}$) | [m] | | $0.1^b$ | 0.25 m regular grid |
| Snow depth ($h_s$) | [m] | | $\dfrac{0.069^c}{\sqrt{N}}$ | 4–5 m spacing, 1.0/2.6 m footprint along-/across-track |
| Snow density ($\rho_s$) | [kg m$^{-3}$] | $300^d$ | $34^e$ | |
| Sea-water density ($\rho_w$) | [kg m$^{-3}$] | $1024^f$ | $0.5^f$ | |
| Sea-ice density ($\rho_i$) | [kg m$^{-3}$] | Eq. (3) | Eq. (4) | |

[a] Pfaffling et al. (2007); Haas et al. (2009); [b] see Sect. 2.3; [c] see Sect. 2.4; Jutila et al. (2021); [d] Warren et al. (1999); [e] King et al. (2020b); [f] Wadhams et al. (1992)

propagation:

$$
\sigma_{\rho_i} = \left[ \left(\frac{\partial \rho_i}{\partial \rho_w}\right)^2 \sigma_{\rho_w}^2 + \left(\frac{\partial \rho_i}{\partial \rho_s}\right)^2 \sigma_{\rho_s}^2 + \left(\frac{\partial \rho_i}{\partial h_{tot}}\right)^2 \sigma_{h_{tot}}^2 + \left(\frac{\partial \rho_i}{\partial h_s}\right)^2 \sigma_{h_s}^2 + \left(\frac{\partial \rho_i}{\partial h_{fs}}\right)^2 \sigma_{h_{fs}}^2 \right]^{\frac{1}{2}}
$$

$$
= \left[ \left(1 + \frac{h_s - h_{fs}}{h_{tot} - h_s}\right)^2 \sigma_{\rho_w}^2 + \left(\frac{h_s}{h_{tot} - h_s}\right)^2 \sigma_{\rho_s}^2 + \left(\frac{\rho_w (h_{fs} - h_s) + \rho_s h_s}{(h_{tot} - h_s)^2}\right)^2 \sigma_{h_{tot}}^2 \right.
$$

$$
\left. + \left(\frac{\rho_w (h_{tot} - h_{fs}) - \rho_s h_s}{(h_{tot} - h_s)^2}\right)^2 \sigma_{h_s}^2 + \left(\frac{\rho_w}{h_{tot} - h_s}\right)^2 \sigma_{h_{fs}}^2 \right]^{\frac{1}{2}}. \tag{4}
$$

Following the uncertainty source analysis of Giles et al. (2007) and using the values summarised in Table 2, the largest contributors to the uncertainty of sea-ice bulk density are, in descending order of magnitude, snow freeboard, snow depth, and total thickness. We disregarded density values with uncertainty exceeding 100 kg m$^{-3}$ from further analysis.

While the assumption of isostatic equilibrium may not necessarily be valid locally, e.g., close to pressure ridges, it holds true when averaging over a sufficient length scale. We varied the averaging length in 10 m increments and found that the mean bulk densities and the standard deviations of the surveys did not change significantly beyond a length scale of about 200 m. Here, we computed sea-ice bulk density estimates at two length scales representing the sensor resolution of the CryoSat-2 satellite as well as the typical resolution of gridded sea-ice thickness data. In the first case, we approximated the scale of full-resolution altimeter footprint by the diameter of a circle with the same area as the 300×1650 m pulse-Doppler-limited footprint of CryoSat-2 in the synthetic aperture radar (SAR) acquisition mode. The diameter of that circle is equal to approximately 800 m. In the second case, we chose the typical satellite product grid cell size of 25 km. We assumed that the sea-ice and snow layers are in isostatic equilibrium at both scales. We calculated an along-track weighted average using the squares of individual





uncertainty values as weights (inverse-variance method):

$$\bar{\rho}_i = \frac{\sum\limits_{i=1}^{N} \frac{1}{\sigma_{\rho_i}^2} \rho_i}{\sum\limits_{i=1}^{N} \frac{1}{\sigma_{\rho_i}^2}} \tag{5}$$

where $N$ equals the number of values within the length scale to be averaged. The resulting uncertainty was determined with

$$\sigma_{\bar{\rho}_i} = \sqrt{\frac{1}{\sum\limits_{i=1}^{N} \frac{1}{\sigma_{\rho_i}^2}}}. \tag{6}$$

We calculated inverse-variance weighted averages also for snow depth since its uncertainty varied spatially. For all other variables we calculated an arithmetic mean.

## 3 Results

This study included a total of 3410 airborne survey kilometres split approximately equally between the years 2017 and 2019 (Table 1). The abundance of different sea-ice types varied between the years and the individual surveys (see the percentages in Table 3). Surveyed sea ice in 2017 was solely FYI. This was a result of the ice-free conditions in the Beaufort and Chukchi Seas in the previous summer, additionally influenced by the collapse of the semipermanent Beaufort high pressure system, and the subsequent reversal of the Beaufort Gyre in the winter prohibiting typical import of MYI to the region (Babb et al., 2020). In 2019, FYI was encountered primarily in the surveys over the southern Beaufort Sea and the Amundsen Gulf but also embedded within the MYI zone in refrozen leads constituting 41 % of the calculated density values. The percentage of SYI was generally low, only around 7 %. The largest percentage of SYI was observed in the survey over the Lincoln Sea on 5 April with minor occurrences on other surveys of that year. Similar to SYI, MYI was included only in the surveys in 2019 not only because of the imported MYI had returned to the Beaufort Sea within the range of the aircraft but also because the data included surveys over the Lincoln Sea and Arctic Ocean within the Last Ice Area where the oldest and thickest sea ice in the Arctic resides (Moore et al., 2019). Overall, approximately half of the sea ice sampled in 2019 was identified as MYI.

Figure 4 shows the backtracked pathways of the sampled sea ice. ICETrack results from 2017 (panel a) confirmed that all sampled ice was FYI. Indicated sea-ice age was up to approximately 180 d in the Beaufort Sea and up to approximately 150 d in the Chuckhi Sea corresponding to freeze-up in October and November, respectively. In 2019 (panel b), the sampled old ice was indeed up to several years old originating from the Beaufort Gyre. The chosen 25 km backtracking segment length did not resolve any SYI in 2019 pointing to a likely very scarce and localised appearance of SYI.

### 3.1 Sea-ice bulk density

Figure 5 shows an example section of a measured sea-ice profile, roughly 30 km in length along a survey track and including different ice types in the Beaufort Sea in 2019. The bulk densities of FYI, SYI, and MYI, derived using the along-track length





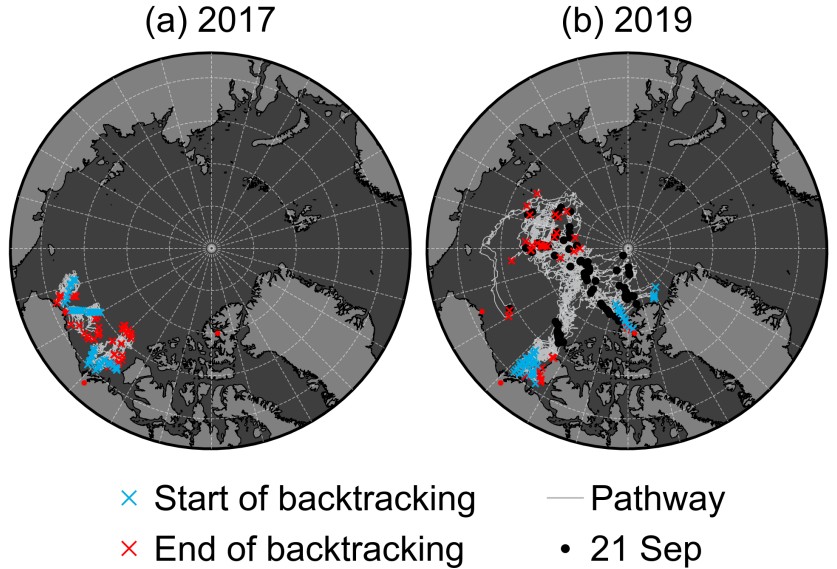

**Figure 4.** Pathways of the sea ice sampled (a) in 2017 and (b) in 2019. Black dots (visible only in (b)) represent the position of the backtracked sea ice on 21 September in the preceding years before sampling, when ice parcels are considered to have survived the summer.

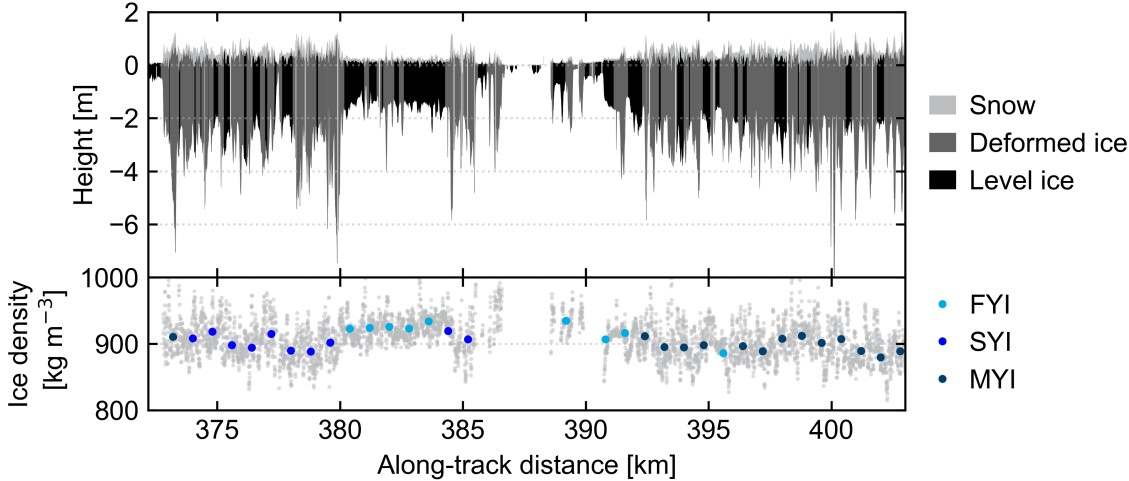

**Figure 5.** Approximately 30 km along-track profile during the survey over the Beaufort Sea on 7 April 2019. The upper panel shows the snow and ice layers, of which the latter is split into deformed and level ice sections, derived from the total thickness, snow freeboard, and snow depth measurements in the native 5–6 m point spacing. The zero height refers to the local sea-surface height. The corresponding calculated sea-ice bulk density values are represented by grey dots in the lower panel together with the 800 m along-track averages that are coded according to the sea-ice type with colour.



**Table 3.** Summary of the 800 m along-track averaged sea-ice bulk density (inverse-variance weighted mean $\pm$ one standard deviation) according to survey and ice type. The mean values correspond to the red crosses in Fig. 6. The percentage of each sea-ice type encountered on each survey is given in parentheses.

| Survey | Region | $\rho_i$ [kg m$^{-3}$] | | | | | |
|---|---|---|---|---|---|---|---|
| | | FYI | | SYI | | MYI | |
| 2 April 2017 | Beaufort Sea | $923.5 \pm 16.0$ | (100 %) | | | | |
| 4 April 2017 | Beaufort Sea | $926.8 \pm 13.9$ | (100 %) | | | | |
| 6 April 2017 | Chukchi Sea | $932.9 \pm 15.3$ | (100 %) | N/A | | N/A | |
| 8 April 2017 | Chukchi Sea | $930.7 \pm 15.8$ | (100 %) | | | | |
| *Mean* | | *$929.3 \pm 16.0$* | *(100 %)* | | | | |
| 2 April 2019 | Lincoln Sea | $873.1 \pm 0.0$ | (0.5 %) | $875.2 \pm 6.9$ | (5.5 %) | $896.0 \pm 19.2$ | (94 %) |
| 5 April 2019 | Arctic Ocean | $931.3 \pm 27.1$ | (2 %) | $907.8 \pm 12.9$ | (23 %) | $907.0 \pm 17.5$ | (75 %) |
| 7 April 2019 | Beaufort Sea | $923.2 \pm 14.8$ | (52 %) | $902.7 \pm 11.4$ | (5 %) | $907.3 \pm 15.5$ | (43 %) |
| 8 April 2019 | Beaufort Sea | $929.7 \pm 17.9$ | (100 %) | N/A | | N/A | |
| 10 April 2019 | Beaufort Sea | $921.6 \pm 18.8$ | (51 %) | $915.6 \pm 10.4$ | (3 %) | $913.6 \pm 17.4$ | (46 %) |
| *Mean* | | *$925.4 \pm 17.7$* | *(41 %)* | *$899.3 \pm 17.4$* | *(7 %)* | *$902.4 \pm 19.4$* | *(52 %)* |
| A10 (Alexandrov et al., 2010) | | $916.7 \pm 35.7$ | | N/A | | $882 \pm 23$ | |

scale of 800 m, are stated in Table 3 and shown in Fig. 6. On average, FYI bulk density was higher than the value derived by Alexandrov et al. (2010) (A10), also for individual surveys. FYI bulk density in 2017 was slightly higher than in 2019, and combined they resulted in an average density of $928.5 \pm 16.4$ kg m$^{-3}$. SYI and MYI bulk densities differed only a little from each other, but were 23–30 kg m$^{-3}$ lower than FYI bulk density. Similar to FYI, the bulk densities of old ice were in the upper
range of or even beyond A10. Figure 7 shows the spatial distribution of the derived sea-ice bulk density when averaged over a typical satellite product grid cell size of 25 km. As expected, lower density values were generally encountered with increasing ice age (Fig. 7e and f). The lowest sea-ice bulk density values were located 50–150 km northwest off the edge of the landfast ice and the coast of Ellesmere Island and Nansen Sound (Fig. 7e). Moreover, there was noticeable along-track variability of bulk density also within a single ice type, even FYI where also the highest bulk density values were observed.

## 3.2 Parametrisation of sea-ice bulk density

We explored the possibility to parametrise sea-ice bulk density using one of the measured sea-ice parameters. Out of the full parameter space, sea-ice freeboard showed the best correlation with the estimated sea-ice bulk density, $r = -0.62$ ($p \ll 0.001$), indicating a significant, linear anti-correlation as expected according to Eq. (2) ($\rho_i \propto -h_{fi}$). However, the dependence on other sea-ice properties and the fact that keels of ridges with high freeboard contain voids filled with sea water introduced



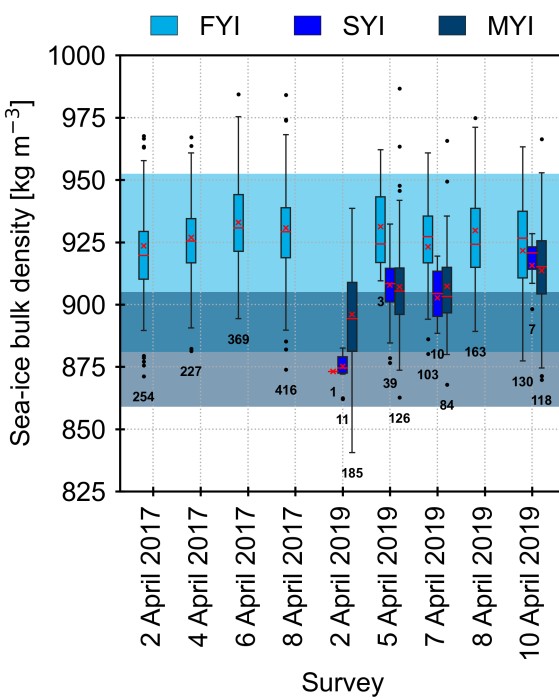

**Figure 6.** Sea-ice bulk density by ice type and survey in the 800 m length scale showing the interquartile range (IQR, Q3–Q1, boxes), the median values (red lines), the inverse-variance weighted mean (red crosses, Table 3), and outliers (beyond 1.5×IQR, dots). The numbers below the boxes correspond to the amount of averaged values contained in each ice type of each individual survey. The shading on the background shows the ± one standard deviation range around the A10 mean FYI (light blue) and MYI (dark blue) densities, respectively.

non-linearity to this relationship. To avoid underestimating the density values near the lower and upper ends of the observed freeboard range and to increase the goodness of fit, we therefore fitted an exponential function to the full, along-track averaged sea-ice bulk density data set ($N = 2246$, Fig. 8). Least-squares fitting yielded a relationship of

$$\rho_i = 72.0 \times e^{-3.74 \times h_{fi}} + 881.8 \tag{7}$$

where sea-ice freeboard is in metres and sea-ice bulk density in kilograms per cubic metre. The resulting coefficient of determination ($R^2$) was 0.42 and RMSE was 15.2 $\mathrm{kg\,m^{-3}}$. The fitted curve showed excellent agreement also with the inverse-variance weighted average densities of the 0.05 m freeboard bins resulting in $R^2 = 0.89$ and RMSE $= 6.2$ $\mathrm{kg\,m^{-3}}$.

## 4 Discussion

We measure sea-ice bulk density indirectly based on (i) the direct measurements of sea-ice thickness, snow depth, and freeboard and (ii) the isostatic balance between the masses of snow, sea ice, and displaced sea water only assuming the densities of the



**Figure 7.** Sea-ice bulk density in 25 km along-track segments shown as coloured circles over the weekly NSIDC EASE-Grid Sea Ice Age product (2–8 April; Tschudi et al., 2019) in bluish colours in 2017 (panels (a) & (c)–(d)) and in 2019 ((b) & (e)–(f)). The close-up panels (c)–(f) are 700×700 km in size and their locations are marked with white squares in the overview panels (a) and (b).

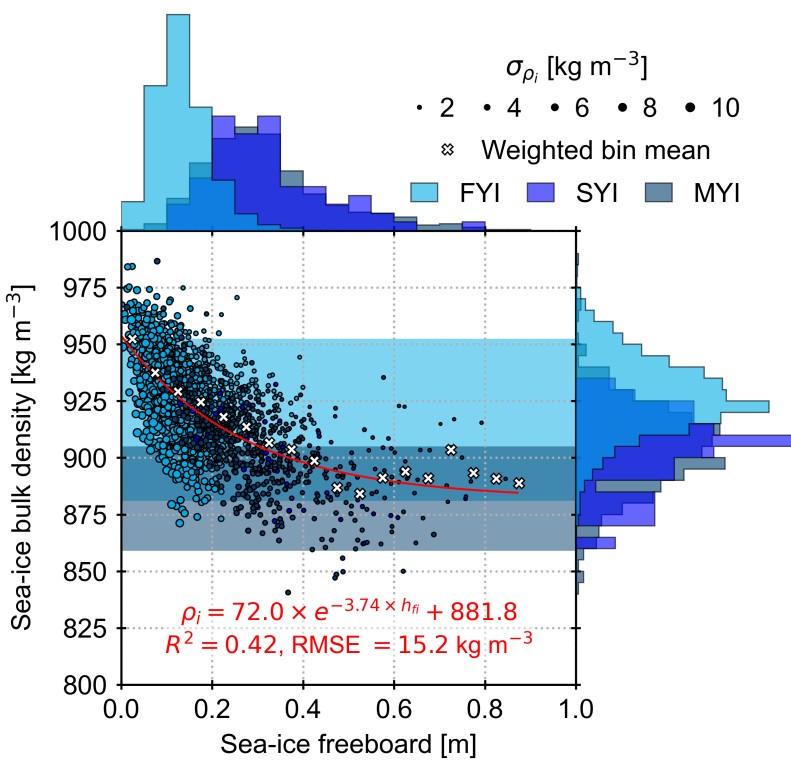

**Figure 8.** Parametrisation of sea-ice bulk density based on sea-ice freeboard. The scatter plot shows all sea-ice bulk density values from 2017 and 2019 in the 800 m length scale ($N = 2246$) against their corresponding sea-ice freeboard values, where the size of the point corresponds to the uncertainty of the density value ($\sigma_{\rho_i}$). The red line and text show the non-linear least-squares fit of an exponential function (Eq. (7)). $R^2$ stands for coefficient of determination and RMSE for root-mean-square error of the fitted curve. The crosses are the inverse-variance weighted means using a sea-ice freeboard bin size of 0.05 m. The shading on the background shows the $\pm$ one standard deviation range around the A10 mean FYI (light blue) and MYI (dark blue) densities, respectively. The histograms on the top and on the right show the probability density functions of freeboard and density, respectively, split into different sea-ice types indicated by colour. The respective bin sizes are 0.05 m and 5 kg m$^{-3}$.

snow layer and sea water. However, we measure the thickness of the sea ice layer and not its mass. Instead, we use sea-ice bulk density to relate sea-ice thickness to the displaced mass of sea water inherently by assuming a constant density of the entire sea-ice layer. However, in reality, the material composition of sea ice is not constant throughout the vertical column which complicates the attribution of bulk sea ice density values to physical properties such as porosity. Above the waterline, the density is lower than that of pure ice due to air incorporated in the pore spaces and to an increasing degree in MYI. Below

the waterline, brine and sea water saturate the sea ice and increase the density above the pure ice density. Despite the indirect measurement method, we are able to detect a difference in FYI bulk density between 2017 and 2019 that can be linked to the high sea-ice deformation in 2017. The effect of deformed and unconsolidated sea-ice is often overlooked and needs the



attention of the scientific community. Dedicated sea-ice porosity studies and extensive field measurement programs, such as the recent Multidisciplinary drifting Observatory for the Study of Arctic Climate (MOSAiC), will be able to shed more light
on their effect and development.

## 4.1 Ice-type averaged sea-ice bulk density

Compared to A10, the average sea-ice bulk density estimates derived in this paper are larger by $11.8\,\mathrm{kg\,m^{-3}}$ ($\approx 1.3\,\%$) for FYI and by $20.4\,\mathrm{kg\,m^{-3}}$ ($\approx 2.3\,\%$) for MYI but still within the A10 uncertainties, albeit close to the upper limit. In general, our ice-type averaged bulk density estimates fall within the range of previous studies (Timco and Frederking, 1996). Reasons
for the comparably high estimates are twofold. First, the A10 FYI density is representative only to level, undeformed ice whereas our estimate includes also deformed FYI. Alexandrov et al. (2010) used drill-hole measurements that were carried out in 3–5 locations, 150–200 m apart around each aircraft landing site on level ice. Sea-water inclusions within deformed and unconsolidated sea ice increase the bulk density. This is a likely reason contributing to the higher FYI bulk density especially in the 2017 data, given the increased deformation caused by the reversal of the Beaufort Gyre. Second, Alexandrov et al. (2010)
calculated the MYI density as a weighted average between the layers above and below the waterline based on values from numerous literature sources but used a density of $550\,\mathrm{kg\,m^{-3}}$ for the upper layer which is significantly lower than the majority of the literature indicates ($720$–$910\,\mathrm{kg\,m^{-3}}$ in Timco and Frederking (1996), $863$–$929\,\mathrm{kg\,m^{-3}}$ in Pustogvar and Kulyakhtin (2016)) and would correspond to an air-volume fraction of up to $40\,\%$. When using the weighted average method in Alexandrov et al. (2010) but the density values from Timco and Frederking (1996) instead, we find a MYI density of $909 \pm 28\,\mathrm{kg\,m^{-3}}$ that
is closer to our estimate.

## 4.2 Uncertainties and limitations of the derived sea-ice bulk density

The effect of the uncertainties in the measured parameters and assumed sea-water and snow density values on the sea-ice bulk density was studied using Gaussian error propagation in Eq. (4). Single point measurements typically resulted in sea-ice bulk density uncertainties of approximately $70\,\mathrm{kg\,m^{-3}}$ and $35\,\mathrm{kg\,m^{-3}}$ for FYI and MYI, respectively. Other sources of
error arise from the different length scales illustrated in Figs. 3 and 5. We are not able to resolve snow depth fully within the total thickness measurement given the comparably large footprint of the EM-Bird but we calculate it as an average of the snow depth measurements along a chord of the circular EM-Bird footprint. Due to the cross-track movement of the EM-Bird under the aircraft, the ground locations and the number of the snow depth measurements within the total thickness measurement vary slightly along the survey track but generally remain at eight to ten measurements close to the centre line.
To ensure representative snow depth estimates, we require at least five valid snow depth measurements for each total thickness measurement, which translates into at least $50\,\%$ coverage along the chord. Errors may occur locally, e.g., at cross-track transition from a sea-ice floe to young ice in a newly refrozen lead as in the leftmost measurement points in Fig. 3, but we assume them to occur randomly and not cause systematic bias. However, uncertainties are reduced when averaged along-track over a length scale. Using the $800\,\mathrm{m}$ length scale, the resulting sea-ice bulk density uncertainty is generally less than $10\,\mathrm{kg\,m^{-3}}$
(Eq. (6)) but remain the highest for thin ice and low sea-ice freeboard where the relative uncertainties of the input parameters





are the largest (see the size of the scatter points in Fig. 8). In turn, averaging over a length scale simplifies the natural variability of sea ice. Figure 5 shows how a single 25 km satellite grid cell can contain already several sea-ice types: level, deformed, FYI, SYI, and MYI. However, assigning the sea-ice types is limited by the spatial (12.5 km) and temporal (weekly) resolution of the NSIDC sea-ice age product, which we try to compensate with the additional thickness-based conditions (see Sect. 2.5.2).

In addition, our measurements are confined to the western Arctic in early April and therefore, more measurements across the Arctic and the seasons are needed to evaluate the spatial and temporal variability of sea-ice bulk density.

### 4.3 Impact on sea-ice thickness retrievals

Assuming all the other parameters for the conversion of freeboard to thickness remain the same, the average sea-ice bulk density estimates derived in this paper would result in 12.4 % and 16.7 % larger sea-ice thickness values for FYI and MYI, respectively,

in comparison to A10. The effect is larger for thicker ice, for which snow depth plays a proportionally less important role. Therefore, improving especially the MYI bulk density is important to derive accurate time series of sea-ice thickness and volume, as in the past, thicker MYI represented a larger fraction of the Arctic sea-ice cover. Kwok and Cunningham (2015) recognised the possibility of varying MYI density between the recent younger MYI and older MYI of the previous decades and discussed the impact of MYI density on sea-ice thickness and volume. Moreover, a potentially increasing degree of deformation

may lead to a bias in the time series, as satellites underestimate sea-ice draft of deformed ice as shown by Belter et al. (2020) and further discussed by Khvorostovsky et al. (2020). Deformed and unconsolidated sea-ice has an increased bulk density due to sea-water inclusions and thus, using a typical density of consolidated level sea-ice for deriving sea-ice thickness from satellite data will eventually lead to an underestimation. Given the thinner and younger sea-ice cover together with observed increase in sea-ice drift speed and deformation (Rampal et al., 2009; Spreen et al., 2011), there is a likely premise for systematic

underestimation by current parametrisations of sea-ice density in regions where and at times when sea ice is deformed. A full impact assessment of the sea-ice bulk density parametrisation on decadal sea-ice thickness data record is a logical next step but beyond the scope of this paper.

### 4.4 Outlook

To represent sea-ice bulk density range as a functional relationship to a parameter observable from space rather than fixed values

based on sea-ice type classification, we parametrised sea-ice bulk density using sea-ice freeboard and obtained a significant correlation. Opting for an exponential function instead of linear was beneficial for ensuring a better fit, capturing the high bulk density values at low freeboard values, and avoiding linear decrease to possibly unrepresentative and unphysical values at high freeboards. Parametrisation in Eq. (7) sets the limits of bulk density to 953.8 $\mathrm{kg\,m^{-3}}$ at zero sea-ice freeboard and approaching 881.8 $\mathrm{kg\,m^{-3}}$ at high freeboards.

Figure 9 shows the parametrisation in Eq. (7) applied to the AWI Level-3 Collocated CryoSat-2 Sea Ice Product (Hendricks and Ricker, 2020) from the winter 2018/2019 converting the monthly gridded sea-ice freeboard to sea-ice density. The resulting sea-ice density distribution had a smoother transition between ice types compared to the current ice-type dependent density classification of the retrieval algorithm. Difference between the density parametrised with Eq. (7) and A10 was positive overall





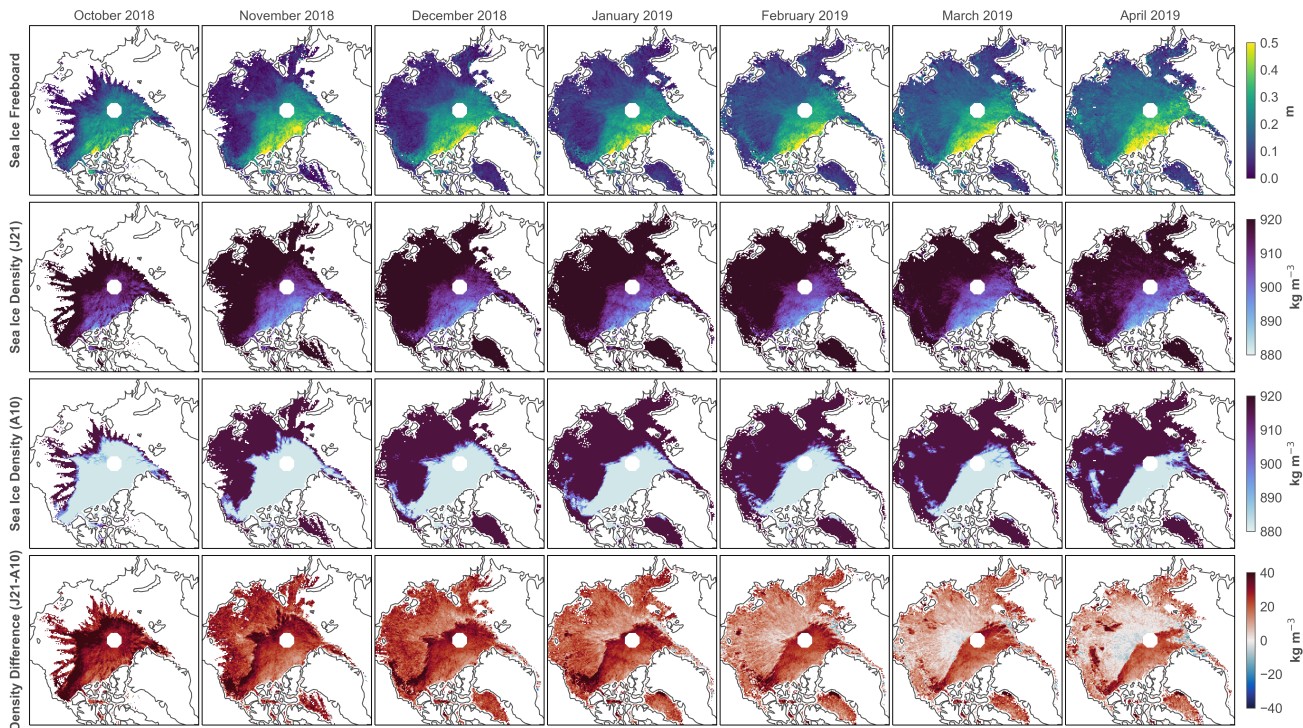

**Figure 9.** Sea-ice density parametrisation applied to the monthly, gridded AWI CryoSat-2 Sea Ice Product (Hendricks and Ricker, 2020) for the winter 2018/2019. The rows show sea-ice freeboard, sea-ice density derived using Eq. (7) (J21), sea-ice density currently used in the product (A10), and the difference between the two (J21−A10). The columns from left to right show monthly means from October 2018 to April 2019.

during the winter except in the Central Arctic Ocean and locally in the Fram and Bering Straits in spring. The density difference
was the largest on MYI in proximity to FYI but decreased toward spring.

    Only less than 3 % of the airborne data set has a sea-ice freeboard value larger than 0.5 m with considerable spread in bulk density values and thus, introducing uncertainty to the parametrisation at high freeboard values. Constraining the parametrisation at high freeboards would require more data in deformed and multi-year sea-ice environments. However, that needs to coincide with a sufficient amount of open leads to ensure accurate conversion of surface elevations to freeboard from the ALS.
With the limitations of the current method, it is also not feasible to investigate cases of negative sea-ice freeboard due to the possible presence of liquid water and altered dielectric properties affecting the retrieval of snow depth.

    Our parametrisation improves upon the previous formulations of sea-ice density given the significantly larger number of data points, larger areal coverage, the variety of ice types including deformed sea ice, and the choice of predictor variable it is based on. Kovacs (1997) based his analysis on 17 FYI and 4 MYI sea-ice cores from the Beaufort Sea and derived a floe-thickness
based non-linear parametrisation of $\rho_i = 936.3 - 18h_i^{0.5}$. Ackley et al. (1976) used drill-hole measurements from 400 m of





profile lines in the Beaufort Sea to derive $\rho_i = -194h'_f + 974$ where $h'_f = h_{fi} + \frac{\bar{\rho}_s}{\bar{\rho}_i}h_s \leq 1.05$ m is effective freeboard and the overbar denotes average density on the floe. The parameters needed for those formulations cannot be directly observed from space, or in the case of effective freeboard not even with a single in situ measurement, which makes them difficult to apply. Demonstrated in Fig. 9, our freeboard-based parametrisation has potential for future applications in satellite altimetry. While

we acknowledge that, due to the variability in snow mass and sea-ice thickness, our parametrisation may not be applicable on sub-kilometre scales as reflected by the scatter in Fig. 8, we think that a sea-ice density parametrisation is a significant improvement upon a single value or fixed values based on ice type. In this paper, we decided to adopt a single-predictor parametrisation for the sake of simplicity. However, for future studies it could be worthwhile, e.g., to apply machine learning algorithms to the full parameter space to discover possible multi-variable relationships. Given the effect of sea-ice deformation

on bulk density, including sea-ice surface roughness, a multi-variable approach could explain more of the variability.

## 5 Conclusions

The unique, collocated, multi-sensor measurements of the Arctic sea ice from the AWI IceBird campaigns allow us not only to observe sea-ice thickness, freeboard, and snow depth in high-resolution on regional scales, but also for the first time to estimate sea-ice bulk densities of different ice types from airborne measurements. Despite measuring the sea-ice bulk density

indirectly by deriving it from other measurements, we are able to capture the effects of deformed ice on FYI bulk density. In the current Arctic, the average FYI and MYI bulk densities are higher than and do not differ as much as earlier studies suggested partly due to including deformed ice in the analysis. Alexandrov et al. (2010) derived a difference of 34.7 $\mathrm{kg\,m^{-3}}$ whereas our measurements show only 26.1 $\mathrm{kg\,m^{-3}}$ providing yet one more indication and consequence that the Arctic sea-ice cover is getting younger. Satellite altimetry sea-ice thickness retrieval algorithms need to adapt to these changes in order

to capture the sea-ice thickness and volume accurately, and to account for changes over the satellite radar altimetry record spanning almost three decades. Taking advantage of the abundant measurements collected over different sea-ice types during two late-winter airborne campaigns, we are able to provide a parametrisation of sea-ice bulk density using sea-ice freeboard. The single-variable exponential function presented here yields a smaller RMSE than the uncertainty of density values fixed by sea-ice type currently in use in large extent. With potential applications in sea-ice thickness retrieval from satellite radar

altimetry, a density parametrisation alone does not completely solve the uncertainty problem in the freeboard-to-thickness conversion. Together with improved knowledge of snow loading, they provide a path to decrease the uncertainty in observing sea-ice thickness and volume where the recent (CryoSat-2/ICESat-2 orbit resonance) and future (CRISTAL mission) advances in dual-altimetry will play a key role. In situ and airborne multi-sensor observations of various sea-ice parameters across the seasons will remain important to validate new approaches.

*Data availability.* A collocated sea-ice parameter data product, including measured total thickness, snow freeboard, snow depth, and surface temperature and derived parameters such as sea-ice thickness, sea-ice freeboard, and sea-ice bulk density, is submitted to and soon available



on PANGAEA (https://www.pangaea.de/). Releasing lower processing-level data is under preparation. Total thickness measurements during the PAMARCMIP2017 campaign are already available at https://doi.org/10.1594/PANGAEA.924848. The EASE-Grid Sea Ice Age, Version 4, is available in the NASA National Snow and Ice Data Center Distributed Active Archive Center: https://doi.org/10.5067/UTAV7490FEPB.
The Canadian Ice Service Arctic Regional Sea Ice Charts in SIGRID-3 Format, Version 1, are available in the NSIDC: National Snow and Ice Data Center: https://doi.org/10.7265/N51V5BW9. The AWI Level-3 Collocated CryoSat-2 Sea Ice Product is available here: ftp://ftp.awi.de/sea_ice/product/cryosat2/v2p3/.

*Author contributions.* AJ carried out the analysis with support from SH and RR and prepared the manuscript with input from all authors. AJ, SH, and RR collected the data. SH applied the density parametrisation to the CryoSat-2 data and produced Fig. 9. LvA processed the
EM-Bird data to identify level and deformed ice. TK conducted the backward-tracking of sea ice with ICETrack.

*Competing interests.* CH is a member of the editorial board of *The Cryosphere*. All other authors declare that they have no conflict of interest.

*Acknowledgements.* The authors thank the crews of *Polar5* and *6*, Kenn Borek Air, AWI technicians and logistics, Environment and Climate Change Canada (ECCC), and Canadian Forces Station (CFS) Alert, who helped at various stages of the data collection. Veit Helm is acknowledged for post-processing the surface temperature data.



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
