# Peer review of "Retrieval and parametrisation of sea-ice bulk density from airborne multi-sensor measurements"

_The Cryosphere, 2021_

## Referee Comment (RC2)

**Review of a manuscript for *The Cryosphere***

Retrieval and parametrisation of sea-ice bulk density from airborne multi-sensor measurements

by A. Jutila et al.

**Overall:**

In this manuscript the authors present the approach to bulk sea ice density retrievals from parallel airborne measurements of total thickness, snow thickness and surface freeboard. For their study the authors use the data retrieved during airborne IceBird campaigns over the Beaufort/Chukchi Sea and Canadian Arctic in the spring seasons of 2017 and 2019.

The authors provide new, generally higher than was used before, estimates for the bulk densities for different types of sea ice. They further propose a new nonlinear parameterization linking observable ice freeboard with sea ice density to be potentially used in satellite based retrievals of sea ice thickness/volume.

The paper is clearly written and results, including figures, are well presented. I therefore consider the manuscript deserves to be published after some moderate modifications according to the comments provided below.

**Major comment:**

My only major comment concerns a new freeboard to density model proposed by the authors. The model is based on exponential fit to the data collected by the authors and offers at the moment RMSE values for the fit itself (model calibration error). However, since the model has a potentially high applicability in the algorithms for ice thickness/ice volume retrievals from satellite-based sensors, it makes sense to have its predictive skills to be tested properly.

Generally, a good agreement with data can be achieved via applying a data model complex enough and hence overfitting; it will not guarantee nevertheless any decent predictive skills for such model.

Since the authors have aggregated a significant volume of measurements for this study, a bootstrapping aproach (or block bootstrapping in case if autocorrelation in the series is substantial) can be used to test the model predicted vs measured values. This routine will provide a more realistic value for the RMSE to be used in future potential uncertainty estimates – RMSE for prediction.

**Other (minor) comments:**

Sec 2.5: "…a sporadically observed by the ALS at fractures (leads) of the seaice cover and we manually selected the corresponding elevations". Is the ALS used onboard receives returns from open water areas too, or the authors refer to refrozen leads only? Would it be possible to use the measured surface temperatures to support the detection of leads? Or this is actually already a part of the procedure for these z-control points identification?

Line 167: Please consider adding a most recent reference to Rosel et al., https://doi.org/10.5194/tc-15-2819-2021; where this effect is also considered.

Line 180: Please clarify the formulation/ application of the level ice criterion. I find it to be not too informative; it is nearly a copypaste from Rabenstein et al which suffice from the same issue.

Lines 185-188: Discussion on age assignment to sea ice along the flight track is somewhat unclear: do the authors refer to an average thickness estimated for level ice only, or for all (level+deformed) ice along a specified transect/transect segment?? If this is the latter, how long the transect segment length used for the age assignment?

Line 238: It can be useful to mention directly (though this is also apparent from eq. 4) that uncertainty in $\sigma_{\rho_i}$ includes spatially variable uncertainty in measured $\sigma_{\rho_s}$, and hence both uncertainties vary along the track.

Table 3:

Table 3 shows numerous numbers with redundant precision in FYI density/density uncertainty estimates. Decimals can be eliminated throughout the table (and the text too in many places) by rounding to the nearest integer to leave significant figures only. E.G. 929.7\pm17.9 -> 930\pm18.

Line 263: typo? "…combined they results…"

Line 289: "… ice due to air incorporated in the pore spaces and to an increasing degree in MYI." Please consider rewriting the sentence. The meaning is clear it only appears awkward.

Line 291: "Despite the indirect measurement method, we are able to detect a difference in FYI bulk density between 2017 and 2019 that can be linked to the high sea-ice deformation in 2017". Please consider referring to Figure 8 here, provided that my comment to figure 8 below is justified."

Figure 8: From the figure it appears that there is a tendency towards higher uncertainties for lower values of density. This is especially clear for FYI where the data may form two groups clusters one below and one smaller group above the fit line. I wonder if these two groups of data points originate from different campaigns? Or could this be only the artefact of the data visualization? This is not to be ruled out (at least for me) as this figure is quite busy. I see no similar tendency for the MYI densities.

---

## Author Response (AR1)

tc-2021-149
**Retrieval and parametrisation of sea-ice bulk density from airborne multi-sensor measurements**

**An item-by-item response to the editor's and referees' comments**

Arttu Jutila et al.

18[th] December, 2021

In the following, you can find *the editor's and referees' comments as a continuous enumerated list in italic* and our responses in regular font in an item-by-item fashion. We refer to the line numbers of the tracked changes document.

**Editor: Michel Tsamados**

1. *Both reviewers are positive but require further work on making the sea ice density analysis more robust. I agree that this dataset and the analysis has the potential to be useful in future parameterizations of sea ice density. I therefore recommend you provide your amended manuscript and for both reviewers to check that they are satisfied with your replies and modifications. I would also like to make a few requests that I feel the reviewers have missed:*

On behalf of all authors, I would like to thank the editor Michel Tsamados for his time and effort in managing our manuscript and for the constructive feedback, which we have considered carefully. We are very grateful for the positive view the editor has and we are confident that with the editor's help the manuscript has improved. I am hopeful that we have been able to meet his expectations and eliminate all his concerns.

2. *Your uncertainty propagation analysis assumes no correlation between variables (in addition to them being Gaussian distributed). This is clearly an incorrect assumptions and if you were to plot H_i vs H_s or H_tot vs H_s you would for sure find strong correlations. Accounting for this will change significantly your uncertainties. See for example Lawrence et al, 2019 for a similar analysis.*

We assume that the editor means the paper of Lawrence et al. (2018) where the uncertainty calculation includes covariance terms. We have considered the uncertainty calculation carefully when conducting the analysis and to our understanding, covariance needs to be considered if there is correlation between the *uncertainties* of the variables, not between the variables themselves (e.g., Taylor, 1997). In our case, the measurements of total thickness, snow depth, and snow freeboard are independent as they arise from observations with separate instruments and thus, the uncertainties are also uncorrelated. On line 235–236, we already state our assumption that the individual uncertainties are uncorrelated. We do not deny that ice and snow thickness would be correlated, but in fact the correlation is complex and not straightforward (e.g., thicker MYI may accumulate more snow than thinner FYI, which in turn hinders further ice growth, but other aspects such as surface roughness

and melting have an impact, too). The main thing here is that, in our case, the related measurement errors are not correlated but independent. The uncertainty of the snow depth measurement from the snow radar has absolutely no connection to the uncertainty of the total thickness measurement from the EM-Bird or to the uncertainty of the snow freeboard measurement from the ALS or vice versa. In the study of Lawrence et al. (2018), however, inclusion of the covariance terms is necessary, because the uncertainties of the satellite radar freeboards affect the respective freeboard correction terms.

> *3. I also second what both reviewer request in motivating more rigorously a one parameter regression and providing a bootstrap type asssesment of your predictive model. Ideally you would like to test your predictions against independent dataset (perhaps the MOSAIC floe?).*

As requested partly by both referees, we have now included more explicit statement on our single-variable parametrisation objective (comment 7) and also a bootstrap approach to evaluate the parametrisation (comment 27). There are no independent sea-ice density observations available to us at the moment.

> *4. I would also like to see more on the length validity of your parameterization. You quickly mention (but don't show) that your results are independent of length scale beyond 200m. Please show this scaling analysis (if anything this could be useful for IS2 application that operate at those length scales). Also, if that is the case why bother with two length scales? And if you analyse both length scales of 800m and 25km why not show these. Please clarify.*

Thank you for pointing this out and asking for clarification. We computed the mean bulk densities for different ice types and for each survey varying the averaging length scale in 10 m increments. We have explained our choice of 800 m on line 245–246 as it corresponds to the footprint size of the CryoSat-2 satellite and relates to our planned further studies and applications. The length scale of 800 m is also a necessary compromise, because the results also depend on the abundance of different ice types, i.e., the number of data points, and for all surveys the 200 m length scale is not representative. In Fig. 1 of this document, we show the length scale analysis for the survey on 10 April 2019 as an example, and it is also evident that FYI bulk density has a larger spread of values than MYI in general. The raw resolution data will be released together with this paper for further studies to examine and apply a length scale most representative to their purposes.

To address this issue, we have expanded the corresponding section in the manuscript, and it reads now (line 245–247): "We varied the averaging length in 10 m increments and found that the mean bulk densities and the standard deviations for different ice types of the surveys did not change significantly beyond a length scale of about 200 m, if measurements of each ice type were abundant."

> *5. Finally, I find your bibliography a bit AWI biased. Please reference earlier work by other groups both historical (i.e. Laxon and Kwok early work) and more recent (i.e. Landy, Qwartly, Lawrence, Garnier, etc...).*

We have considered your suggestion thoroughly and decided to include Quartly et al. (2019) into our reference list. For the other references suggested, we have trouble to connect them to our manuscript. The named first authors have publications almost exclusively about satellite altimetry and not about airborne measurements of sea ice, which is the topic of our paper. Their works also barely discuss sea-ice density beyond naming it as a source of uncertainty, which further reduces the significance to our paper. Kwok and Cunningham (2015), which is already included in our paper (line 362), refer to their own earlier work in Kwok and Cunningham (2008) and also to previous sea-ice density parametrisations in Ackley et al. (1976) and Kovacs (1997). In their review paper, Quartly et al. (2019) underline the obvious demand for enhanced in situ observations of sea-ice density (and snow depth) to update current parametrisations, therefore we have included it in the introduction (line 51), but they do not provide any solutions. Out of the 62 references in the now revised manuscript, only about one quarter of them has a lead author affiliated to AWI and in total of about one third has at least one AWI author. Clear majority of the references are not related to AWI.

[Figure]

**Figure 1:** Example of the length scale analysis of the survey on 10 April 2019.

**Anonymous Referee #1**

6. *This paper addresses a critical parameter of Arctic sea ice needed for the determination of sea ice thickness for satellite altimetry, and is timely considering the recent launch of ICESat-2. The results determined here are very likely to be widely used and be impactful. The authors have a tremendous data set to examine sea ice density, and the analysis is thorough and well described. The paper is well written and figures are of high quality. My comments are primarily on one point – the authors simplify all their analysis to a single empirical fit, while I believe it would be more useful to explore the variability of this relationship for different ice types/conditions in more detail. Based on what they show, I think this would be straightforward to do without too much effort. I recommend publication after my comments below are addressed.*

On behalf of all authors, I would like to thank the referee for their time and effort in reviewing our manuscript and for the constructive feedback, which we have considered carefully. We are very grateful for the very positive evaluation by the referee, and we are confident that with the referee's help the manuscript has improved. I am hopeful that we have been able to meet their expectations and eliminate all their concerns.

**Major comments**

7. *You went to a lot of work to examine the relationship between sea ice density and different ice types, and deformed vs level. You also indicate you looked at the relationship between density and other parameters besides ice freeboard. But in the end chose to present only a single relationship based on ice freeboard. This seems somewhat unsatisfying, given you do show that there are systematic differences and density for FYI and MYI, and it appears there may be a difference between level and deformed ice (judging from figure 4 – it would be nice to also have a plot of density vs deformation; or if just two categories, a plot of the density distribution for these two categories so the reader could tell if that was a significant difference or not). My guess is your work is going to be very highly cited and this relationship will be used for almost all future altimetric estimates of Arctic ice thickness, so this will have a big influence. It would be nice if it could either be refined a bit better, or shown that such refinement results in no significant difference. So, I would have liked to see this relationship (eq 7), presented for just FYI, and just MYI, and if possible, just deformed and just level. It would be really interesting to see if that makes*

*any difference, or it's just within the bounds of the error. Perhaps the difference is not big enough to matter, maybe because the ice freeboard captures a lot of the variability inherent in these different ice classes. If so, that is worth reporting, because that will save future authors from trying it, or even provide some more guidance on the kind of observations are needed to improve things more. I think this could be done with a quite modest amount of effort, since you have already identified which ice is in which class.*

Our objective, perhaps not communicated well enough, was to find a simple, single-variable, functional relationship between sea-ice bulk density, including deformed ice, and a parameter observable from space. As this is the first study with this data set, we wanted to keep the parametrisation simple and provide one, good, all-around tool instead of up to half a dozen equations depending on the ice type (FYI/SYI/MYI+level/deformed). A single-variable parametrisation is directly comparable to the existing density parametrisations in Ackley et al. (1976) and Kovacs (1997) (line 394–397). To address the issue raised by the referee, we have been more explicit about our objective at the end of the introduction section when describing it in the revised manuscript (line 89–90). Further investigations on possible relationships with other or multiple parameters and more advanced parametrisations will be a topic of a future study, as proposed at the end of discussion (line 403–405). The full resolution data will be made public to the scientific community together with this paper.

The referee asks for a figure of density vs. deformation or density distributions of level and deformed ice. However, we cannot provide this with the along-track averaged (800 m) data, where such a length scale can already include a mixture of ice types and data points are not classified as level and deformed anymore. That is also evident from Fig. 5, to which we think the referee is referring instead of Fig. 4 (backtracking). We would be extremely cautious to use the non-averaged data to analyse this, as the assumption of isostatic equilibrium may not be valid locally at the nominal resolution, especially for deformed ice. However, to answer the referee's request, we provide such plots only for this response document in Fig. 2 below. It can be seen that the distributions of level and deformed ice densities have similar shapes with mean values differing 5–6 kg m$^{-3}$, whereas deformation increases the spread of values only in sea-ice freeboard.

The referee also asks to see the parametrisation of Eq. (7) for just FYI and just MYI, which we provide here in Fig. 3 of this document below. It is clear that the parametrisation split to different ice types leads to coefficient of determination ($R^2$) values worse than for the complete data set. We have reported this in the revised manuscript (line 304–305) but keep the single parametrisation for the complete data set.

8. *You also mention that for your fit you tried other parameters and they didn't have good correlations. That's good to know, but maybe provide more details? What parameters exactly, and how poor were the fits? Would a multiple regression that included more variables improve things. For example, would including ice freeboard and ice type improve it much, or not?*

The other available parameters were total thickness, sea-ice thickness, snow depth, snow freeboard, and surface temperature, each including also their minimum, maximum, and standard deviation values. None of them showed significant linear, exponential, or power law dependency to density. For example, Eq. (2) would imply also linear anti-correlation between density and snow depth, but the result was an obscure cloud of data points with a correlation coefficient of only −0.33 compared to −0.62 of sea-ice freeboard. We have now mentioned this on line 304–305.

We agree with the referee that multiple regression has potential to improve the parametrisation. Therefore, at the very end of the discussion section (line 403–405), we propose future studies to apply multi-variable approaches and machine learning to explain more of the variability in density.

9. *If indeed your relationship is the best, and trying other fancier parameterizations doesn't make much of a difference, then as I say, this is the one thing from your paper that everyone will use. In that case, maybe it is worth putting this relationship in the abstract itself? Ok, maybe interested*

[Figure]

**Figure 2:** The nominal resolution, i.e., non-averaged, data showing the density distributions of level (blue) and deformed (orange) ice in the FYI (top) and old ice (bottom) regimes. The horizontal dashed lines over the histograms show the respective mean values indicated in the legends in the lower right corners with standard deviation. PDF stands for probability density function.

[Figure]

**Figure 3:** Density parametrisation of Eq. (7) and Fig. 8 of the manuscript split into different ice types and fits: FYI exponential (left) and linear (middle) fit and old ice (SYI+MYI) exponential fit (right).

> *readers shouldn't be so lazy.*

While we agree that the parametrisation is a key result of the paper, putting it in the abstract would require adding also explanations of the variables used as required by the journal's instructions. Our current abstract is already at the journal's upper limit of length and therefore, we refrain from adding it to the abstract. As also pointed out by the referee, we hope that interested readers can find it from the text or in Fig. 8.

> *10. Section 2.6 – you do a nice job of accounting for the uncertainties. But it seems like you are*

*assuming they are all normally distributed here. But you noted a bias in the snow radar; maybe there are biases in the other measurements, too (e.g. a bias in the EM-bird for ridges). Did you correct the data for any of these biases so that the errors would be centered first? Another possible bias is suggested from the retrieval rates in table 1. Do you know if there is any ice types or thickness for which retrievals are less likely? I am thinking mostly of the snow radar, which I believe will get poor retrievals for thin snow, and possibly also in heavily deformed ice. This doesn't bias your data exactly, because this is excluded, but it may bias the types of ice that you measure (i.e. your data might not be an average representation for the whole Arctic, or even for your survey areas). Thus, your density fit might be biased to certain ice types. It would also be nice to have some more discussion and analysis of whether this relationship would have more error in different regions or ice types and conditions (this relates to the main points above about the simplified empirical fit).*

We did not correct the data for any biases but assumed that the errors are normally distributed and uncorrelated. The mean bias of 0.86 cm in the snow radar is below the sensor resolution and within the accuracy of the ground truth data (line 168–169), on which this value is based, and thus, negligible (Jutila et al., 2021). Regarding the snow radar retrievals, please see our response below to the referee comment 18 about Section 2.4.

Regarding the retrieval rates of Table 1, data gaps exist due to the following main reasons:

- EM-Bird (total thickness)
  - Brief ascents every 15-20 minutes to monitor the sensor drift during post-processing (line 121)
- ALS (snow freeboard)
  - No freeboard conversion before (after) first (last) lead tie point, especially over landfast sea-ice (line 138–139)
  - For example, the beginning of the survey on 2 April 2019 is over the landfast sea-ice of the Nansen Sound.
- Snow Radar (snow depth)
  - EM-Bird calibration ascents
  - Anomalously low retrieval rate on 7 April 2019 is due to a momentary malfunction of the instrument
  - Characteristics of the snow depth retrieval algorithm, see below our response to the referee comment 18 about Section 2.4

Therefore, we agree with the referee that our density fit is somewhat biased as it does not include landfast sea-ice. Other than that no specific ice type is excluded in our analysis.

**Minor comments:**

11. *Line 3 "in the 1980s and earlier" I think reads a bit better.*

Thank you for the suggestion. We agree and have corrected it.

12. *Line 25 Perhaps change "Coming to the era of" to "At the start of the era of"*

We have replaced it with "At the beginning of the era of" to avoid repeating the word "start" later in the sentence.

13. *Line 35-44 – note that W99 was updated by Webster et al, and Blanchard-Wrigglesworth et al examined the spatial bias as well. Though not sure if these are the updates you are referring to, but you should probably provide a cite for the updated product, and one or more of the reanalysis techniques.*

Since the focus of this paper is not on the different snow depth products, we decided to refer to the inter-comparison study of Zhou et al. (2021) to avoid adding considerable length to the introduction by including a comprehensive list of citations. If the referee means the snow reconstruction study of Blanchard-Wrigglesworth et al. (2018), it is also included in the paper of Zhou et al. We do not see it fair to mention and thus highlight only one updated product or reanalysis technique study.

To address this issue, we have moved the following sentence earlier in the text and added more explicit expressions to lead the reader to the descriptions first (line 40): "Descriptions of the different snow depth products currently available can be found in the inter-comparison study of Zhou et al. (2021, **and references therein**) **and in broad outlines below**."

We do not agree with the referee's note about W99 being updated by Webster et al., presumably referring to the paper Webster et al. (2014) describing the interdecadal changes in snow depth. To our understanding, Webster et al. studied the change in regional springtime snow depth by comparing W99 and airborne snow radar measurements from Operation IceBridge 2009–2013 interpolated using the same two-dimensional quadratic method as in W99. They provide estimates of how much the snow cover has been thinning, partly in keeping with the results of Kurtz and Farrell (2011), but that is hardly a snow depth product in the same sense as the ones included in Zhou et al.

14. *Lines 50-64 – It may see obvious, but perhaps point out here that you are focusing on Arctic sea ice density. There have been a few studies that measured Antarctic sea ice density, which because of different properties may be expected to have different densities and effective densities (though they tend to span the same range as these Arctic observations).*

While the review articles on sea-ice density we refer to (Timco and Frederking, 1996; Timco and Weeks, 2010) do not distinguish between Arctic and Antarctic sea ice, we agree with the referee that it is important to point out that majority of the measurements originate from Arctic sites. We have added the following sentences to the end of the respective paragraph (line 65): "It has to be noted that majority of the density measurements originate from Arctic sites, which is the case also in our study. Different properties and processes of Antarctic sea ice could lead to different densities values."

In connection to this, we noticed that the abstract did not mention the geographical location of our study. We have added this to the revised version of the manuscript (see below the section Additional corrections by authors).

15. *Line 121 – How often do you get total thickness less than snow freeboard or snow depth? This is obviously a measurement error, so makes sense to exclude. But does it tell you something about your measurement error? i.e. when this happens you are getting an ice thickness error of 100%, or, based on buoyancy, an error in ice thickness something like at least 3 times the snow freeboard. Is it possible you also have errors of this magnitude in the other direction (i.e. grossly overestimating ice thickness)?*

Total thickness less than snow freeboard or snow depth occurs only when total thickness is close to or below 0.1 m, i.e., the accuracy of the EM-Bird instrument, or when calculated sea-ice freeboard is negative. These data points comprise less than 0.3 % of the entire data set, but they are disregarded through filtering before analysis. In general, these measurements have the largest relative uncertainties. We do not think that our approach could lead to overestimated ice thickness nor have we observed any such indications.

16. *Line 129-130 – Can you give a bit more detail? What is a typical spacing of these sea surface references? I am assuming it's pretty small so that linear interpolations between them works just fine.*

The spacing between leads is diverse and depends on the ice regime. The typical spacing is up to 10 km. Over FYI in the Beaufort Sea and Chukchi Sea, spacing is often below 10 km. In contrast, the MYI north of the Canadian Archipelago is densely packed and the spacing can be more than 30 km.

We apply a spline interpolation between the tie points. Data before the first lead detection and after the last lead detection during a survey flight are discarded to avoid extrapolation errors.

We have elaborated the corresponding section of the manuscript (line 135–139). Please note also the related comment 28 from the referee #2.

17. *Line 146 – "in snow depth"*

Typically, radar range resolution is given in relation to a medium in which the radar wave is propagating, such as, free-space/air or snow. Here, we refer to range resolution in the said medium, snow, and not in the measured parameter. To make it clearer, we have changed the expression "range resolution was 1.14 cm in snow" to "range resolution in snow was 1.14 cm" on line 154.

18. *Section 2.4 – different snow depth retrieval algorithms for ultrawideband radar have been tried, with differing results. I see you did some validation of your method and report errors, so that is good. Can you add a comment on how well your algorithm is expected to do versus others? It may be that yours works well enough for the ice type you validated against, but perhaps it might have larger errors elsewhere?*

As we state on line 158–160, the workflow and the retrieval algorithm are described in detail in our recent publication Jutila et al. (2021). There we compared our peakiness algorithm against the wavelet-based algorithm of Newman et al. (2014). We found that due to the characteristics of the wavelet method (detected interfaces are both on the leading edge of the radar waveform) it was prone to both overestimation and underestimation of snow depth compared to our algorithm and in situ data. To our knowledge, other retrieval algorithms are not publicly available as open source for comparison.

Another aspect to consider, in addition to the retrieval algorithm, is the CReSIS Snow Radar itself. Since its first deployment on the Operation IceBridge campaign in 2009, the radar has been continuously developed (e.g., Yan et al., 2017; Arnold et al., 2020; MacGregor et al., 2021). The lack of stability in radar properties and design over the years has somewhat hampered developing algorithms but in turn revealed differences in radar return power and sidelobe levels between campaigns (Kurtz et al., 2013; Kwok and Haas, 2015; Kwok et al., 2017). What makes the comparison against our study difficult is the fact that our radar version is similar to the latest one used on Operation IceBridge, but that specific version is not used anywhere else.

We agree with the referee that our retrieval algorithm was validated only for a specific ice type, namely level and landfast FYI. Increase of errors could be expected over rougher sea-ice surfaces, but we are confident that our approach is valid also in such sea-ice environments for two reasons. First, the low altitude of our IceBird surveys results in a radar footprint with an approximately 2 m diameter that is only 3–7 % in size of the earlier high-altitude acquisitions (line 154–157). This considerably smaller radar footprint size decreases the amount and possibility of off-nadir reflections that could lead to potentially erroneous snow depth retrievals. Second, we require that a surface topography ("roughness") estimate is $h_{topo} \leq 0.5$ m within the radar footprint to filter out retrievals over very rough surfaces where our method is not validated (line 164–167). In a single survey, less than 2 % of the values are disregarded due to this threshold. In a small area such as the radar footprint, surfaces rougher than the used threshold value correspond to sharp sails of pressure ridges, which are often snow-free due to wind erosion.

Further validation opportunities against in situ measurements on other ice types are not available to this date, because they were not realised due to poor weather (campaign in 2019) or cancelled altogether due to the global COVID-19 pandemic (campaign in 2021). However, we will pursue further validation opportunities over a range of different sea-ice surfaces in future campaigns.

19. *Section 2.6 I am a little confused on how these uncertainties come into the final analysis. It looks like in Table 3 and figures 6 and 8, you just use the standard deviation, so in the end these apriori uncertainties go away. I think this is actually ok if the uncertainties are normally distributed,*

*but not if there are biases (such as in snow depth). I gather from section 4.2 that equation (4) is used to calculate the local (800m or 25 km) densities and their respective uncertainties (eq. 5 and 6). But then I assume these do not affect the values in Table 3 or the empirical fit? Or were the uncertainties explicitly used in the fitting procedure? Granted, they are quite small relative to the scatter, so I think they wouldn't affect the fit at all. I do see the discussion in section 4.2, so perhaps all that is needed is a sentence in section 2.6 to clarify how they are used.*

We think this confusion stems from us using the term "inverse-variance weighted mean" to describe the along-track averaging method, when we in fact mean that we used the inverse of squared individual *uncertainties* as weights. We apologise for this mix-up of terms and we have made appropriate corrections for the revised manuscript. All mean sea-ice density values presented in this study — averaged over a length scale, survey, or ice type — are weighted with uncertainties according to Eqs. (5) and (6). Therefore, the uncertainty information is not lost but included in the averaging.

As explained in the figure caption, Fig. 6 features interquartile range, which is not equal to standard deviation. Fig. 8 shows local (800 m, Eq. (6)) uncertainties as explained in the figure caption. In Table 3, we report standard deviation, as explained in the table caption, to enable direct comparison with the values in Alexandrov et al. (2010). The local (800 m) densities have respective uncertainty values that are used for the uncertainty-weighted averaging of the survey or the entire ice type. Therefore, sea-ice density uncertainties are used in calculating the uncertainty-weighted means according to Eq. (5) again and not lost. Maybe that is the missing sentence from Section 2.6. More confusion is perhaps created as the ice type averaged density mentioned in Table 3 is not an arithmetic mean calculated from the individual survey means but again an uncertainty-weighted mean of all 800 m averaged density measurements of the ice type in question. The uncertainties of the local (800 m) densities are not used in finding the exponential parametrisation, because they did not significantly affect the fit as the referee already pointed out.

20. *Figure 5 – this figure is what really makes me want to see the differences in the distribution of density for deformed vs undeformed and whether there is a statistically significant difference. I understand that you probably couldn't do it in 800 m along track averages.*

That is correct, we don't distinguish between level and deformed ice after averaging over the 800 m length scale. Please see our response to the referee's major comment 7 above.

21. *Figure 8 – actually, relating to my top comment, this figure does show quite well the difference in density distributions for different ice types. I suspect that there is no statistical significant difference between the SYI and MYI distributions, but probably there is with FYI. Most of the FYI have low sea ice freeboard, so maybe the relationship is just us good if it was based on ice type? I suppose sea ice freeboard could be capturing that ice type relationship, but as I noted above, you might not get sea ice freeboard from altimetry, but you might get e.g. roughness.*

Yes, Fig. 8 shows density distributions for FYI, SYI, and MYI. We don't understand the referee's comment about a relationship based on ice type, which is not a continuous parameter and does not provide a functional relationship. We provide average density estimates based on ice type in Table 3. Lastly, while not all altimeters are able to measure sea-ice freeboard, it can be derived from Ku band radar altimeters, which can be found onboard the current CryoSat-2 or the future CRISTAL satellite missions. In contrast, laser altimeters, such as onboard the current ICESat-2 satellite, are not able to measure sea-ice freeboard but snow freeboard. For them, surface roughness could potentially reveal a relationship to density, especially between level and deformed ice (line 404–405).

22. *Line 290-292 – This again argues for showing a relationship between density and deformation.*

We cannot show the relationship with the 800 m averaged data, as the referee already noted above. Before averaging, the assumption of isostatic equilibrium may not be valid. Please see our response to the referee's major comment 7 above.

23. *Equation 7 and figure 8 – you should state the uncertainty in the fit parameters in equation 7, and maybe show the confidence limits of the fit on figure 8.*

We agree with the referee and we have added the requested information to the revised manuscript (line 298–300 and Fig. 8). The parameters of the exponential fit $\rho_i = a \times e^{b \times h_{fi}} + c$ are the following: $a = 72.0 \pm 2.4\,\text{kg m}^{-3}$, $b = -3.7 \pm 0.4\,\text{m}^{-1}$, and $c = 881.8 \pm 3.1\,\text{kg m}^{-3}$. The 95 % confidence band of the fit is updated and shown in Fig. 4 of this response document below.

[Figure]

**Figure 4:** Figure 8 of the manuscript updated with the 95 % confidence band of the fit (red shading) and the new formulation of the parametrisation.

24. *Discussion/Conclusions –Your data are for April only. People will be tempted to use your relationship generally, which as you note might not be so valid (or over other areas, too). It is worth stressing this as a caution to users. Can you also speculate, based on your data and the literature, how the results might be different elsewhere or at other times? e.g. could it be that in the autumn densities might be higher because of saltier FYI and maybe less consolidated ridges? Do you think the scatter in the fit in figure 8 would capture the range of densities likely to be observed elsewhere and at other times?*

Thank you for raising this valid point. We agree with the referee that our data originates only from April and it is also regionally restricted. We discuss it on line 354–356 together with other limitations and uncertainties and we have stressed it further where appropriate in the revised manuscript. We further agree that it is worthwhile to emphasise, as the currently widely used density values derived in Alexandrov et al. (2010) share the same limitations. Their FYI measurements come from the airborne Sever expeditions that "took place mainly from mid March to early May, when landing on ice floes was possible. Thus, the data represent late winter conditions before melting starts." In addition, their FYI measurements are from level ice only. Despite these limitations, they are currently widely in use by various satellite altimetry algorithms across seasons and the Arctic regions (e.g., Sallila et al., 2019; Quartly et al., 2019).

Regarding the speculation on differing results elsewhere and at other times, we share the referee's thoughts on denser ice in newly formed and more saline FYI as well as unconsolidated ridges. Timco

and Frederking (1996) report a range of density values of 720–940 kg m$^{-3}$ that includes also our observations. Compared to Timco and Frederking, our density range is missing the lowest values that originate perhaps from brine-drained or rotten summer sea-ice measurements. However, we would prefer to base our discussion on actual measurements and results.

25. *Author contributions – contribution of CH is not specified.*

We will include the contribution of CH explicitly as follows: "AJ [- -] prepared the manuscript with input from SH, RR, LvA, TK, and CH."

**Referee #2: Dmitry Divine**

**Overall:**

26. *In this manuscript the authors present the approach to bulk sea ice density retrievals from parallel airborne measurements of total thickness, snow thickness and surface freeboard. For their study the authors use the data retrieved during airborne IceBird campaigns over the Beaufort/Chukchi Sea and Canadian Arctic in the spring seasons of 2017 and 2019.*

    *The authors provide new, generally higher than was used before, estimates for the bulk densities for different types of sea ice. They further propose a new nonlinear parameterization linking observable ice freeboard with sea ice density to be potentially used in satellite based retrievals of sea ice thickness/volume.*

    *The paper is clearly written and results, including figures, are well presented. I therefore consider the manuscript deserves to be published after some moderate modifications according to the comments provided below.*

On behalf of all authors, I would like to thank referee Dmitry Divine for his time and effort in reviewing our manuscript and for the constructive feedback, which we have considered carefully. We are very grateful for the very positive evaluation by the referee, and we are confident that with the referee's help the manuscript has improved. I am hopeful that we have been able to meet his expectations and eliminate all his concerns.

**Major comment:**

27. *My only major comment concerns a new freeboard to density model proposed by the authors. The model is based on exponential fit to the data collected by the authors and offers at the moment RMSE values for the fit itself (model calibration error). However, since the model has a potentially high applicability in the algorithms for ice thickness/ice volume retrievals from satellite-based sensors, it makes sense to have its predictive skills to be tested properly.*

    *Generally, a good agreement with data can be achieved via applying a data model complex enough and hence overfitting; it will not guarantee nevertheless any decent predictive skills for such model.*

    *Since the authors have aggregated a significant volume of measurements for this study, a bootstrapping aproach (or block bootstrapping in case if autocorrelation in the series is substantial) can be used to test the model predicted vs measured values. This routine will provide a more realistic value for the RMSE to be used in future potential uncertainty estimates – RMSE for prediction.*

We have evaluated the RMSE value of our exponential parametrisation using the suggested bootstrapping approach with $10^4$ random samples of the measurements, which resulted in an average

RMSE of 15.2 kg m$^{-3}$ with a 95 % confidence band of 14.8–15.7 kg m$^{-3}$. The average RMSE is equal to previously reported RMSE, but we have reported the increased accuracy of the RMSE and include the confidence band in the revised manuscript (line 299–300).

**Other (minor) comments:**

28. *Sec 2.5: "...a sporadically observed by the ALS at fractures (leads) of the seaice cover and we manually selected the corresponding elevations". Is the ALS used onboard receives returns from open water areas too, or the authors refer to refrozen leads only? Would it be possible to use the measured surface temperatures to support the detection of leads? Or this is actually already a part of the procedure for these z-control points identification?*

Yes, the ALS receives returns also from open water close to the nadir. Open water targets away from the centre line tend to reflect the laser beam away. The sea-surface height tie points are selected manually from both open water and newly refrozen leads with negligible sea-ice freeboard, preferably from the centre line beam with the strongest laser returns. Because the tie points are selected manually, utilising surface temperature data has not been necessary at this point. Perhaps an automatic lead detection scheme using surface temperature data can be developed in future.

We have elaborated the corresponding section of the manuscript (line 133–134). Please note also the related comment 16 from the referee #1.

29. *Line 167: Please consider adding a most recent reference to Rosel et al., https://doi.org/10.5194/tc-15-2819-2021; where this effect is also considered.*

Thank you for the suggestion. Rösel et al. (2021) was published after the submission of our manuscript and we have added it for the revised version. However, it must be noted that Rösel et al. as well as Kurtz and Farrell (2011) and Kurtz et al. (2013) use radar versions and retrieval algorithms different to ours which may limit the direct applicability of their results (line 176–179).

30. *Line 180: Please clarify the formulation/ application of the level ice criterion. I find it to be not too informative; it is nearly a copypaste from Rabenstein et al which suffice from the same issue.*

We thank the referee for asking for details. We have extended the description of the level ice criterion in the revised manuscript as follows (line 191–199): "We started with identifying level and deformed ice following the approach of Rabenstein et al. (2010). The filter is based on the observation that level ice is mostly flat and extends over long distances. We identified data points that fulfilled those characteristics using two criteria. First, we calculated the along-track total thickness gradient using a three-point Lagrangian interpolator. We applied a threshold gradient of 4 cm within an along-track distance of 1 m, below which the ice was classified as level following Rabenstein et al. (2010). Second, this condition must be met continuously for at least 100 m of the profile length. Choosing the value of 100 m, which represents approximately twice the footprint size of the EM-bird, makes sure that the conditions were met over two completely independent EM total thickness measurements. If these criteria were not fulfilled, the ice was deemed deformed."

31. *Lines 185-188: Discussion on age assignment to sea ice along the flight track is somewhat unclear: do the authors refer to an average thickness estimated for level ice only, or for all (level+deformed) ice along a specified transect/transect segment?? If this is the latter, how long the transect segment length used for the age assignment?*

The sea-ice age (years) is assigned by collocating the EM-Bird measurement locations at the nominal resolution of 5-6 m sample spacing with the NSIDC product or Canadian Ice Service charts. This does not depend on ice deformation. To take into account the spatial (12.5 km grid) and temporal (weekly) resolution of the sea-ice age products and possible sea-ice drift, we finally define sea-ice type (FYI/SYI/MYI) according to sea-ice age and thickness together. This way we aim to avoid some of the potentially erroneous classifications. For example, assigned age of 1 year but level thickness

of 3 m indicates thermodynamically grown MYI and not FYI as the sea-ice age would simply first suggest. Similarly, all individual measurements with sea-ice thickness less than 2 m are classified as FYI. When considering the data that has been along-track averaged over a length scale, we consider the mean sea-ice thickness including both level and deformed ice within each length scale (800 m or 25 km).

32. *Line 238: It can be useful to mention directly (though this is also apparent from eq. 4) that uncertainty in \sigma_rho_i includes spatially variable uncertainty in measured \sigma_rho_s, and hence both uncertainties vary along the track.*

The uncertainty of snow density $\sigma_{\rho_s}$ was not measured but taken as constant (Table 2), so we assume that the referee really means the uncertainty of snow depth $\sigma_{h_s}$ in his comment. However, as the referee points out, the factors influencing the uncertainty of sea-ice density are already apparent in Eq. (4).

33. *Table 3: Table 3 shows numerous numbers with redundant precision in FYI density/density uncertainty estimates. Decimals can be eliminated throughout the table (and the text too in many places) by rounding to the nearest integer to leave significant figures only. E.G. 929.7\pm17.9 -> 930\pm18.*

We prefer to keep the precision in FYI density estimates and their standard deviation values in Table 3 to ease the direct comparison with the values in Alexandrov et al. (2010). In the final data product, the density values are released with corresponding uncertainty values that inform the potential user of the precision.

34. *Line 263: typo? "...combined they results..."*

We refrain from any modifications for two reasons. First, the subject and the verb must agree in number, i.e., either both singular or plural. Second, results are generally written in past tense, because they refer to completed work. Here, we meant that measurements over FYI in both 2017 and 2019 put together resulted in an average density of $928.5 \pm 16.4 \, \mathrm{kg \, m^{-3}}$. Therefore, we have kept the expression "...combined they resulted in...".

35. *Line 289: "... ice due to air incorporated in the pore spaces and to an increasing degree in MYI." Please consider rewriting the sentence. The meaning is clear it only appears awkward.*

We have split the sentence into two parts (line 313–314): "Above the waterline, the density is lower than that of pure ice due to air incorporated in the pore spaces. This feature is more pronounced in MYI."

36. *Line 291: "Despite the indirect measurement method, we are able to detect a difference in FYI bulk density between 2017 and 2019 that can be linked to the high sea-ice deformation in 2017". Please consider referring to Figure 8 here, provided that my comment to figure 8 below is justified."*

Please see our response to the comment 37 below.

37. *Figure 8: From the figure it appears that there is a tendency towards higher uncertainties for lower values of density. This is especially clear for FYI where the data may form two groups clusters one below and one smaller group above the fit line. I wonder if these two groups of data points originate from different campaigns? Or could this be only the artefact of the data visualization? This is not to be ruled out (at least for me) as this figure is quite busy. I see no similar tendency for the MYI densities.*

The points raised by the referee are indeed artefacts in the data visualisation. Admittedly, Figure 8 is busy, that is the unfortunate downside of the significant data volume (more than 2000 data points).

There is no tendency toward higher uncertainty for lower density values but there is for lower sea-ice freeboard as we state on line 348–350. That is related to all measurements (total thickness, snow depth, and snow freeboard) having relatively small values and thus, large relative uncertainties. The FYI data points from the campaigns in 2017 and 2019 overlap with each other and do not form distinct clusters on either side of the fit line.

**Additional corrections by authors**

In the abstract, we have added a description of the study area to complement the study period (line 7): "Our sea-ice density measurements are based on over 3000 km of high-resolution collocated airborne sea-ice and snow thickness and freeboard measurements **in the western Arctic Ocean** in 2017 and 2019."

In Eq. (5) and (6), the subscript $i$ is repeated as the index and as the abbreviation for ice. Therefore, to avoid confusion, we have modified the equations by changing the index symbol to $k$.

In the reference list, the IEEE Early Access information of Jutila et al. (2021) has been updated with volume number (60) and article sequence number (4300716) and year (2022) following publication. The citation is now:
Jutila, A., King, J., Paden, J., Ricker, R., Hendricks, S., Polashenski, C., Helm, V., Binder, T., and Haas, C.: High-Resolution Snow Depth on Arctic Sea Ice From Low-Altitude Airborne Microwave Radar Data, IEEE Transactions on Geoscience and Remote Sensing, 60, 4300 716, https://doi.org/10.1109/TGRS.2021.3063756, 2022

**Data availability**

The data products related to this paper have been prepared by PANGAEA and they are ready to be published together with the paper. Corresponding DOIs have been updated to the revised manuscript.

- Jutila, A., Hendricks, S., Ricker, R., von Albedyll, L., Haas, C.: Airborne sea ice parameters during the PAMARCMIP2017 campaign in the Arctic Ocean, Version 1, *PANGAEA*, https://doi.pangaea.de/10.1594/PANGAEA.933883, in review, 2021.

- Jutila, A., Hendricks, S., Ricker, R., von Albedyll, L., Haas, C.: Airborne sea ice parameters during the IceBird Winter 2019 campaign in the Arctic Ocean, Version 1, *PANGAEA*, https://doi.pangaea.de/10.1594/PANGAEA.933912, in review, 2021.

**References**

Ackley, S. F., Hibler, W. D., Kugzruk, F. K., Kovacs, A., and Weeks, W. F.: Thickness and roughness variations of Arctic multiyear sea ice, Tech. Rep. 76-18, Cold Regions Research and Engineering Laboratory, 1976.

Alexandrov, V., Sandven, S., Wahlin, J., and Johannessen, O. M.: The relation between sea ice thickness and freeboard in the Arctic, The Cryosphere, 4, 373–380, https://doi.org/10.5194/tc-4-373-2010, 2010.

Arnold, E., Leuschen, C., Rodriguez-Morales, F., Li, J., Paden, J., Hale, R., and Keshmiri, S.: CReSIS airborne radars and platforms for ice and snow sounding, Annals of Glaciology, 61, 58–67, https://doi.org/10.1017/AOG.2019.37, 2020.

Blanchard-Wrigglesworth, E., Webster, M. A., Farrell, S. L., and Bitz, C. M.: Reconstruction of Snow on Arctic Sea Ice, Journal of Geophysical Research: Oceans, 123, 3588–3602, https://doi.org/10.1002/2017JC013364, 2018.

Jutila, A., King, J., Paden, J., Ricker, R., Hendricks, S., Polashenski, C., Helm, V., Binder, T., and Haas, C.: High-Resolution Snow Depth on Arctic Sea Ice From Low-Altitude Airborne Microwave Radar Data, IEEE Transactions on Geoscience and Remote Sensing, pp. 1–16, https://doi.org/10.1109/TGRS.2021.3063756, 2021.

Kovacs, A.: Estimating the full-scale flexural and compressive strength of first-year sea ice, Journal of Geophysical Research: Oceans, 102, 8681–8689, https://doi.org/10.1029/96JC02738, 1997.

Kurtz, N. T. and Farrell, S. L.: Large-scale surveys of snow depth on Arctic sea ice from Operation Ice-Bridge, Geophysical Research Letters, 38, L20 505, https://doi.org/10.1029/2011GL049216, 2011.

Kurtz, N. T., Farrell, S. L., Studinger, M., Galin, N., Harbeck, J. P., Lindsay, R., Onana, V. D., Panzer, B., and Sonntag, J. G.: Sea ice thickness, freeboard, and snow depth products from Operation IceBridge airborne data, The Cryosphere, 7, 1035–1056, https://doi.org/10.5194/tc-7-1035-2013, 2013.

Kwok, R. and Cunningham, G. F.: ICESat over Arctic sea ice: Estimation of snow depth and ice thickness, Journal of Geophysical Research, 113, C08 010, https://doi.org/10.1029/2008JC004753, 2008.

Kwok, R. and Cunningham, G. F.: Variability of Arctic sea ice thickness and volume from CryoSat-2, Philosophical Transactions of the Royal Society A: Mathematical, Physical and Engineering Sciences, 373, 20140 157, https://doi.org/10.1098/rsta.2014.0157, 2015.

Kwok, R. and Haas, C.: Effects of radar side-lobes on snow depth retrievals from Operation IceBridge, Journal of Glaciology, 61, 576–584, https://doi.org/10.3189/2015JoG14J229, 2015.

Kwok, R., Kurtz, N. T., Brucker, L., Ivanoff, A., Newman, T., Farrell, S. L., King, J., Howell, S., Webster, M. A., Paden, J., Leuschen, C., MacGregor, J. A., Richter-Menge, J., Harbeck, J., and Tschudi, M.: Intercomparison of snow depth retrievals over Arctic sea ice from radar data acquired by Operation IceBridge, The Cryosphere, 11, 2571–2593, https://doi.org/10.5194/tc-11-2571-2017, 2017.

Lawrence, I. R., Tsamados, M. C., Stroeve, J. C., Armitage, T. W. K., and Ridout, A. L.: Estimating snow depth over Arctic sea ice from calibrated dual-frequency radar freeboards, The Cryosphere, 12, 3551–3564, https://doi.org/10.5194/tc-12-3551-2018, 2018.

MacGregor, J. A., Boisvert, L. N., Medley, B., Petty, A. A., Harbeck, J. P., Bell, R. E., Blair, J. B., Blanchard-Wrigglesworth, E., Buckley, E. M., Christoffersen, M. S., Cochran, J. R., Csathó, B. M., De Marco, E. L., Dominguez, R. T., Fahnestock, M. A., Farrell, S. L., Gogineni, S. P., Greenbaum, J. S., Hansen, C. M., Hofton, M. A., Holt, J. W., Jezek, K. C., Koenig, L. S., Kurtz, N. T., Kwok, R., Larsen, C. F., Leuschen, C. J., Locke, C. D., Manizade, S. S., Martin, S., Neumann, T. A., Nowicki, S. M., Paden, J. D., Richter-Menge, J. A., Rignot, E. J., Rodríguez-Morales, F., Siegfried, M. R., Smith, B. E., Sonntag, J. G., Studinger, M., Tinto, K. J., Truffer, M., Wagner, T. P., Woods, J. E., Young, D. A., and Yungel, J. K.: The scientific legacy of NASA's Operation IceBridge, Reviews of Geophysics, 59, e2020RG000 712, https://doi.org/10.1029/2020RG000712, 2021.

Newman, T., Farrell, S. L., Richter-Menge, J., Connor, L. N., Kurtz, N. T., Elder, B. C., and McAdoo, D.: Assessment of radar-derived snow depth over Arctic sea ice, J. Geophys. Res. Ocean., 119, 8578–8602, https://doi.org/10.1002/2014JC010284, 2014.

Quartly, G. D., Rinne, E., Passaro, M., Andersen, O. B., Dinardo, S., Fleury, S., Guillot, A., Hendricks, S., Kurekin, A. A., Müller, F. L., Ricker, R., Skourup, H., and Tsamados, M.: Retrieving Sea Level and Freeboard in the Arctic: A Review of Current Radar Altimetry Methodologies and Future Perspectives, Remote Sensing, 11, 881, https://doi.org/10.3390/rs11070881, 2019.

Rabenstein, L., Hendricks, S., Martin, T., Pfaffhuber, A., and Haas, C.: Thickness and surface-properties of different sea-ice regimes within the Arctic Trans Polar Drift: Data from summers

2001, 2004 and 2007, Journal of Geophysical Research, 115, C12 059, https://doi.org/10.1029/2009JC005846, 2010.

Rösel, A., Farrell, S. L., Nandan, V., Richter-Menge, J., Spreen, G., Divine, D. V., Steer, A., Gallet, J.-C., and Gerland, S.: Implications of surface flooding on airborne estimates of snow depth on sea ice, The Cryosphere, 15, 2819–2833, https://doi.org/10.5194/tc-15-2819-2021, 2021.

Sallila, H., Farrell, S. L., McCurry, J., and Rinne, E.: Assessment of contemporary satellite sea ice thickness products for Arctic sea ice, The Cryosphere, 13, 1187–1213, https://doi.org/10.5194/tc-13-1187-2019, 2019.

Taylor, J. R.: An Introduction to Error Analysis: The Study of Uncertainties in Physical Measurements, chap. 9. Covariance and Correlation, pp. 209–226, University Science Books, Sausalito, California, 2nd edn., 1997.

Timco, G. W. and Frederking, R. M.: A review of sea ice density, Cold Regions Science and Technology, 24, 1–6, https://doi.org/10.1016/0165-232X(95)00007-X, 1996.

Timco, G. W. and Weeks, W. F.: A review of the engineering properties of sea ice, Cold Regions Science and Technology, 60, 107–129, https://doi.org/10.1016/j.coldregions.2009.10.003, 2010.

Webster, M. A., Rigor, I. G., Nghiem, S. V., Kurtz, N. T., Farrell, S. L., Perovich, D. K., and Sturm, M.: Interdecadal changes in snow depth on Arctic sea ice, Journal of Geophysical Research: Oceans, 119, 5395–5406, https://doi.org/10.1002/2014JC009985, 2014.

Yan, J.-B., Gogineni, S., Rodriguez-Morales, F., Gomez-Garcia, D., Paden, J., Li, J., Leuschen, C. J., Braaten, D. A., Richter-Menge, J. A., Farrell, S. L., Brozena, J., and Hale, R. D.: Airborne Measurements of Snow Thickness: Using ultrawide-band frequency-modulated-continuous-wave radars, IEEE Geoscience and Remote Sensing Magazine, 5, 57–76, https://doi.org/10.1109/MGRS.2017.2663325, 2017.

Zhou, L., Stroeve, J., Xu, S., Petty, A., Tilling, R., Winstrup, M., Rostosky, P., Lawrence, I. R., Liston, G. E., Ridout, A., Tsamados, M., and Nandan, V.: Inter-comparison of snow depth over Arctic sea ice from reanalysis reconstructions and satellite retrieval, The Cryosphere, 15, 345–367, https://doi.org/10.5194/tc-15-345-2021, 2021.

---

## Author Response (AR2)

tc-2021-149

**Retrieval and parametrisation of sea-ice bulk density from airborne multi-sensor measurements**

**An item-by-item response to the editor's decision**

Arttu Jutila et al.

22$^{\text{nd}}$ December, 2021

In the following, you can find *the editor's decision as a bullet point list in italic* and our responses in regular font in an item-by-item fashion. We refer to the line numbers of the revised tracked changes document.

**Editor: Michel Tsamados**

- *Dear authors*
  *I am happy that you have addressed all the referees comments as well as my additional request for clarification (you convinced me about the error covariances).*

We are very pleased that our responses and revision have been satisfactory.

- *I encourage the authors to check that all variables in equation (4) have been clearly defined (and double check about the lack of correlation of their uncertainties). Could you prove that point?*

The definitions of the variables in Eq. (4) are described in the respective preceding subsections of the Section 2, Data and methods. Additionally, they are summarised in Table 2 and part of them also graphically in Fig. 1 in the manuscript.

- *I still think that the paper by Landy et al (2020) where the authors explicitly look at the relative contribution of sea ice density to the total thickness estimates (their Figure 8) is relevant to this paper (at least as an example of the potential application to satellite processing chains and as it uses the exact same equation). I leave it to the authors to decide.*
  *Landy, J. C., Petty, A. A., Tsamados, M., & Stroeve, J. C. (2020). Sea ice roughness overlooked as a key source of uncertainty in CryoSat-2 ice freeboard retrievals. Journal of Geophysical Research: Oceans, 125(5), e2019JC015820.*

We had indeed overlooked Landy et al. (2020) as it focuses on sea-ice roughness, but as the editor rightfully points out, it also discusses the effect of other component uncertainties including sea-ice density. Therefore, we have decided to include this to our reference list and mention it in the very beginning of the introduction (line 23).

**Additional edits by authors**

**Data availability**

Following the acceptance of the manuscript, the related data sets have been released to the public in PANGAEA and their DOIs have been registered. The DOIs have been updated to the revised manuscript and the full citations are the following:

- Jutila, A., Hendricks, S., Ricker, R., von Albedyll, L., Haas, C.: Airborne sea ice parameters during the PAMARCMIP2017 campaign in the Arctic Ocean, Version 1, *PANGAEA*, https://doi.org/10.1594/PANGAEA.933883, 2021.

- Jutila, A., Hendricks, S., Ricker, R., von Albedyll, L., Haas, C.: Airborne sea ice parameters during the IceBird Winter 2019 campaign in the Arctic Ocean, Version 1, *PANGAEA*, https://doi.org/10.1594/PANGAEA.933912, 2021.

**Acknowledgements**

We have added a short statement to acknowledge the efforts of the referees and the editor.

**References**

[revised manuscript text omitted]